# Development and Biological Evaluation of the First Highly Potent and Specific Benzamide-Based Radiotracer [^18^F]BA3 for Imaging of Histone Deacetylases 1 and 2 in Brain

**DOI:** 10.3390/ph15030324

**Published:** 2022-03-08

**Authors:** Oliver Clauß, Linda Schäker-Hübner, Barbara Wenzel, Magali Toussaint, Winnie Deuther-Conrad, Daniel Gündel, Rodrigo Teodoro, Sladjana Dukić-Stefanović, Friedrich-Alexander Ludwig, Klaus Kopka, Peter Brust, Finn K. Hansen, Matthias Scheunemann

**Affiliations:** 1Department of Neuroradiopharmaceuticals, Institute of Radiopharmaceutical Cancer Research, Research Site Leipzig, Helmholtz-Zentrum Dresden-Rossendorf, 04318 Leipzig, Germany; b.wenzel@hzdr.de (B.W.); m.toussaint@hzdr.de (M.T.); w.deuther-conrad@hzdr.de (W.D.-C.); d.guendel@hzdr.de (D.G.); r.teodoro@life-mi.com (R.T.); s.dukic-stefanovic@hzdr.de (S.D.-S.); f.ludwig@hzdr.de (F.-A.L.); k.kopka@hzdr.de (K.K.); p.brust@hzdr.de (P.B.); 2Pharmaceutical and Cell Biological Chemistry, Pharmaceutical Institute, University of Bonn, 53121 Bonn, Germany; l.schaeker@uni-bonn.de (L.S.-H.); finn.hansen@uni-bonn.de (F.K.H.); 3Institute for Drug Discovery, Medical Faculty, Leipzig University, 04103 Leipzig, Germany; 4Faculty of Chemistry and Food Chemistry, School of Science, Technical University Dresden, 01062 Dresden, Germany

**Keywords:** histone deacetylase inhibitor, HDAC1/2-specific, radiochemistry, fluorine-18 labelling, positron emission tomography (PET), brain-penetration, glioblastoma, glioma

## Abstract

The degree of acetylation of lysine residues on histones influences the accessibility of DNA and, furthermore, the gene expression. Histone deacetylases (HDACs) are overexpressed in various tumour diseases, resulting in the interest in HDAC inhibitors for cancer therapy. The aim of this work is the development of a novel ^18^F-labelled HDAC1/2-specific inhibitor with a benzamide-based zinc-binding group to visualize these enzymes in brain tumours by positron emission tomography (PET). **BA3**, exhibiting high inhibitory potency for HDAC1 (IC_50_ = 4.8 nM) and HDAC2 (IC_50_ = 39.9 nM), and specificity towards HDAC3 and HDAC6 (specificity ratios >230 and >2080, respectively), was selected for radiofluorination. The two-step one-pot radiosynthesis of [^18^F]**BA3** was performed in a TRACERlab FX2 N radiosynthesizer by a nucleophilic aliphatic substitution reaction. The automated radiosynthesis of [^18^F]**BA3** resulted in a radiochemical yield of 1%, a radiochemical purity of >96% and a molar activity between 21 and 51 GBq/µmol (*n* = 5, EOS). For the characterization of **BA3**, in vitro and in vivo experiments were carried out. The results of these pharmacological and pharmacokinetic studies indicate a suitable inhibitory potency of **BA3,** whereas the applicability for non-invasive imaging of HDAC1/2 by PET requires further optimization of the properties of this compound.

## 1. Introduction

Epigenetics describe inheritable and reversible regulation mechanisms of genes expression without altering the DNA sequence. It plays a pivotal role in gene expression by enzyme-mediated post-translational modifications (PTMs) of proteins, such as formylation, methylation, ubiquitination and acetylation. Two of the most studied modifications are the processes of deacetylation and acetylation of lysine side chains on histones, which are regulated by the opposing enzymes histone deacetylases (HDACs) and histone acetyltransferases (HATs). The HDAC family consists of 18 isoforms, which are subdivided into four classes. The enzymes of classes I, II and IV are zinc-dependent enzymes, whereas class III enzymes, also known as sirtuins, are NAD^+^-dependent (Table 1) [1,2,3]. The enzymatic deacetylation of lysine residues by HDACs leads to a positive charge on histones, and consequently modulates the chromatin structure by increasing the interaction of histones with the negatively charged phosphate groups of the DNA. The condensed chromatin structure restricts access of transcription factors to the DNA, and thus results in transcriptional repression. Conversely, the acetylation of histones by HATs is associated with a more open, and relaxed chromatin conformation and induced transcriptional activation. Over the past years, accumulating evidence showed that dysregulations of the histone deacetylation mechanism impact, among others, cell proliferation, apoptosis, cell differentiation and inflammation processes. These are associated with the progression of pathophysiological states, such as cancer, neurodegenerative disorders or memory impairment [2,3,4,5]. Furthermore, the upregulation of HDACs1/2 was discovered in several types of cancers, including gastric, colorectal, and prostate cancer as well as glioblastoma and several haematological malignancies. Both HDACs share a sequence similarity of around 93% and are usually co-expressed in several large co-repressor complexes, such as NuRD and MiDAC [6,7,8,9]. HDACs1/2 were also found to play a critical role in tumourigenesis and are inversely correlated with disease-free intervals and overall survival [4,10,11,12,13,14,15]. Given their regulatory activity in multiple processes involved in cancer progression and resistance to radiotherapy and chemotherapy, the development of HDAC-targeted therapies using single agents or drug combinations is very attractive for tumour entities presenting a poor prognosis, and treatment resistance such as glioblastoma [16,17,18]. For these reasons, the use of HDAC inhibitors in the management of gliomas is studied in clinical trials using pan-inhibitors: **valproic acid** (NCT00302159), **panobinostat** (NCT00859222) or **vorinostat** (NCT01236560) as adjuvant therapy to radiotherapy and/or chemotherapy. However, the development of such a personalized therapy would benefit from a non-invasive imaging tool to help (1) assessing the drug targeting potential and (2) to gain more fundamental understanding of the HDAC1/2 density and role in healthy and pathogenic brain tissue, which is still not well understood.

Therefore, we aim to develop a radiotracer for the non-invasive imaging of HDACs1/2 in the brain via positron emission tomography (PET). The evaluation of the expression and distribution patterns in vivo will help to understand the significance of these enzymes in neuronal processes and pathologies such as glioblastoma.

In the last two decades, numerous HDAC inhibitors and radiotracers have been developed, but only a few were able to cross the blood–brain barrier (BBB) and exhibit suitable pharmacokinetics for brain imaging. HDAC inhibitors contain generally a zinc-binding group (ZBG), a linker and a cap group, visualized in Figure 1 on the structures of published inhibitors. Over the years hydroxamic acids and benzamides have emerged as the most frequently used ZBGs.

Hydroxamic-acid-based radiotracers such as [^18^F]**SAHA** [19] and [^18^F]**FAHA** [20,21,22,23,24,25], structurally based on the FDA-approved drug **vorinostat** (**SAHA**) [26], showed either poor BBB penetration ([^18^F]**SAHA**) or acted as a substrate for HDACs and thus was metabolized rapidly in vivo ([^18^F]**FAHA**). Characterized as pan-HDAC inhibitors, these compounds notoriously lack selectivity towards specific HDAC isoforms or classes. In 2014, the cinnamoyl hydroxamic acid [^11^C]**martinostat** was developed as an HDAC1, 2, 3 and 6-specific radiotracer with good BBB penetration [27,28,29,30]. In contrast, benzamide-based inhibitors such as **tacedinaline** and **entinostat** (see Figure 1) generally demonstrate selectivity towards class I HDACs and do not show genotoxicity [31,32,33,34]. Consequently, radiolabelled benzamide inhibitors have emerged as an alternative to hydroxamic acids for the diagnostic imaging of class I HDACs in brain via PET. Nevertheless, there are only a few benzamide-based radiotracers published. Additionally, both the ^11^C-radiolabelled HDAC1-3-specific inhibitors **tacedinaline** [^11^C]**CI-994** [35] and **entinostat** [^11^C]**MS-275** [36] showed a low brain uptake in vivo. However, while developing a series of benzamide-based brain-penetrable radiotracers, it was discovered that a benzylic amine, e.g., in [^11^C]**I** (Figure 1), is responsible for an improved brain uptake [35]. Furthermore, the specificity towards HDAC1, as well as HDAC2, can be improved by a substitution on the 5-position of the 2-aminoanilide structure (see Figure 1) with a (hetero)aromatic moiety. Such bulky, lipophilic moieties are able to engage the so-called “foot pocket”, an internal cavity in these enzymes [37,38,39,40,41].

The aim of this work is the development of a novel ^18^F-labelled HDAC1/2-specific inhibitor with a benzamide-based ZBG for targeting these enzymes in brain tumours. Therefore, a series of fluorinated compounds, based on **entinostat** and **tacedinaline**, bearing bulky, lipophilic moieties on the 5-position of the *ortho*-aminoanilide scaffold, to increase the specificity towards the HDACs1/2, was designed. Subsequently, the designed compounds were synthesized and their inhibitory potencies against HDAC1, 2, 3 and 6 were determined. The most potent compound was radiofluorinated and biologically evaluated regarding its in vitro cell toxicity in different brain tumour cell lines as well as its in vivo metabolism and pharmacokinetics.

## 2. Results and Discussion

### 2.1. Organic Chemistry

A small library of new compounds (**BA1**–**BA10**) was synthesized using a six-step synthesis route (Figure 1). As fluorine-bearing moiety the 2-fluoropropanamide was used instead of the apparently obvious fluoroacetamide moiety. The [^18^F]fluoroacetamido group is known to suffer from limited stability in vivo and to carry the risk of defluorination [22,23,42,43]. Both processes would be disadvantageous for imaging by PET. In contrast, the 2-[^18^F]fluoropropanoyl group is used frequently for PET tracer design, due to higher metabolic stability [44,45,46,47]. Additionally, it is expected to have a low impact on biological activity due to its small size.

Starting from 4-bromo-2-nitroaniline, the Boc protection of the aromatic amino group was performed in two steps [48]. First of all, the amino group was treated with di-*tert*-butyl dicarbonate (Boc_2_O) and catalytic amounts of *N*,*N*-dimethyl-4-aminopyridine (DMAP) to afford the bis-Boc-protected intermediate followed by cleavage of one Boc group using trifluoroacetic acid (TFA) to give *tert*-butyl (4-bromo-2-nitrophenyl)carbamate (**1**) in a total yield of 95%. Under these conditions it was possible to achieve higher yields with this two-step synthesis compared with the reported yields [49,50,51,52,53] using a one-step synthesis. Subsequently, **1** was reacted with 2-/3-thienyl-, 2-/3-furanyl- or *para*-fluoro-phenylboronic acid in a Suzuki cross coupling reaction with tetrakis(triphenylphosphine)palladium(0) (Pd(PPh_3_)_4_) in 1,4-dioxane/water (7:3, *v*/*v*) to provide the corresponding compounds **2a**–**e** in very good yields (90–96%). In the first approach, the reduction in the aromatic nitro group was carried out under hydrogen atmosphere with palladium on carbon and afforded **3a**–**c** in quantitative yields [49]. Using the same procedure for the 3-furan-containing compound **2e**, the desired aniline product could not be obtained. The NMR analysis showed that both, the aromatic nitro group and the 3-furan, were entirely reduced to form the corresponding aniline derivative substituted with 3-tetrahydrofuran instead of 3-furan. Therefore, a reduction by sodium dithionite was carried out as a milder alternative, which gave the desired products **3d** and **3e** in good to moderate yields (83% and 40%, respectively) [54]. Subsequently, the amide couplings of **3a** and **3b** with 4-(Fmoc-aminomethyl)benzoic acid were performed in a straight forward synthesis under standard conditions using (benzotriazol-1-yloxy)tripyrrolidinophosphonium hexafluorophosphate (PyBOP) as coupling reagent. After cleaving the Fmoc protecting group with a solution of 20% piperidine in *N*,*N*-dimethylformamide (DMF), the resulting amines were coupled with 2-fluoropropanoic acid under the same reaction conditions and led to the compounds **4a** and **4b** (23% and 40%, respectively, over three steps). To bypass the use of protecting groups and increase the yield of the amide coupling, we synthesized the 4-[(2-fluoropropanamido)methyl]- and 4-(2-fluoropropanamido)benzoic acids **7** and **8** by acylation of the amino group of 4-(aminomethyl)-(PAMBA) and 4-aminobenzoic acid (PABA) with 2-fluoropropanoyl chloride **6** in good yields (63% and 79%, respectively). With the compounds **7** and **8** we were then able to perform the final amide coupling in one instead of three reaction steps to synthesize the corresponding compounds **4a**–**e** and **5a**–**e** in better yields (57–95%). Finally, the Boc protecting group was removed with a solution of 30% TFA in dichloromethane (DCM) to give the final products **BA1–BA10** after precipitation in moderate to good yields (15–91%).

### 2.2. Determination of the In Vitro Inhibitory Potency

All prepared compounds as well as the reference HDAC inhibitors **vorinostat**, **entinostat** and **tacedinaline** were tested regarding their inhibitory activity against the class I enzymes HDAC1-3. Additionally, HDAC6 was used as a control isoform to evaluate the selectivity towards class IIb HDACs. The IC_50_ values are shown in Table 2.

The HDAC inhibitors containing (hetero)aryl substituted benzamides as ZBG showed, as expected, a high inhibitory activity against HDAC1 with IC_50_ values ranging from 4.8 nM to 24.2 nM for **BA1**–**BA10**. In contrast, negligibly low inhibitions of HDAC3 and HDAC6 were observed. The screening also revealed that the compounds with the PAMBA linker (*n* = 1, **BA1**/**BA3**/**BA5**/**BA7**), compared with the PABA linker (*n* = 0, **BA2**/**BA4**/**BA6**/**BA8**), have a higher inhibitory potency towards HDAC1. The only exceptions are **BA9** and **BA10**, which both bear a 3-furan substitution and exhibit no substantial differences in the IC_50_ values against HDAC1. Compared with the approved hydroxamic acid-based HDAC inhibitor **vorinostat** and the benzamides **tacedinaline** and **entinostat**, the inhibitory potencies of the new compounds are significantly improved towards HDAC1. It is known that the use of a foot pocket-targeting (hetero)aromatic substituent at position 5 of the aminoanilide moiety leads to an increase in the specificity for HDAC1 and HDAC2 over HDAC3 [37,38,39,40,41]. This could be proven by the specificity ratios (HDAC1/HDAC3) determined for the synthesized compounds ranging from >74 (**BA6**) to 479 (**BA1**) compared with nearly no specificity of the reference compounds **vorinostat**, **entinostat** or **tacedinaline**, respectively. Out of the series, **BA1** and **BA3** showed the best inhibitory activities against HDAC1. **BA3** was selected for radiolabelling as the most active inhibitor of HDAC1 and HDAC2.

### 2.3. Precursor- and Radiosynthesis

#### 2.3.1. Manual Radiosynthesis

For a direct radiofluorination of [^18^F]**BA3**, the corresponding Boc-protected bromine precursor **9** was synthesized as shown in Figure 1 in an overall yield of 34%. The radiolabelling process was investigated under thermal heating using 2–3 mg of precursor **9** and different (i) polar aprotic solvents, (ii) bases, (iii) different precursor to base ratios, (iv) temperatures, (v) reaction times and (vi) azeotropically dried ^18^F-fluorination agents. First, the conventional K[^18^F]F-K_2.2.2_ was used for radiofluorination of the intermediate Boc-[^18^F]**BA3** in DMF, dimethyl sulfoxide (DMSO) and acetonitrile (MeCN) as solvents. The reactions were carried out at 150 °C (DMF/DMSO) or 100 °C (MeCN) for up to 20 min. Aliquots of the reaction mixtures were taken every 5 min and analysed by radio-thin layer chromatography (radio-TLC) and/or radio-reversed phase high-performance liquid chromatography (radio-RP-HPLC) to determine the radiochemical yield (RCY) of the labelling process. In DMF or DMSO with K_2_CO_3_ as base, the radiosynthesis of Boc-[^18^F]**BA3** was not possible. According to radio-RP-HPLC, under these reaction conditions a polar radioactive by-product was formed in high amounts (30% in DMF, 45% in DMSO). When using MeCN as a solvent, it was possible to obtain Boc-[^18^F]**BA3** in labelling yields between 1% and 2% (with KHCO_3_ or K_2_CO_3_, respectively), but the same radioactive by-product (~10%) was formed as shown by radio-RP-HPLC. The use of K_2_C_2_O_4_ instead of KHCO_3_/K_2_CO_3_ and 18-crown-6 instead of K_2.2.2_ did not result in an increase in the radiochemical yield of Boc-[^18^F]**BA3**. Finally, the additional impact of microwave irradiation, which was investigated in DMSO, DMF and MeCN, respectively, did not improve the radiofluorination. For all syntheses, the (radio-)RP-HPLC chromatograms showed a decomposition of precursor **9** and Boc-[^18^F]**BA3** already after 5 min, as well as a large amount of unreacted [^18^F]fluoride.

In general, these conditions are strongly basic, limiting the synthetic utility for base-sensitive precursors since various side reactions, such as eliminations, can occur [56]. Assuming that the reaction conditions described above are not suitable for radiosynthesis of Boc-[^18^F]**BA3**, the use of [^18^F]tetra-*n*-butylammonium fluoride ([^18^F]TBAF) was investigated as a milder reagent [57]. The impact of (i) different solvents, (ii) temperatures, and (iii) reaction times were studied. The radiofluorination of Boc-[^18^F]**BA3** was investigated first in MeCN at 100 °C achieving labelling yields up to 7% after 15 min, which were not increasing with further reaction time. Then, the influence of *tert*-butanol (*tert*-BuOH) as a solvent at 100 °C was investigated and a labelling yield of 5% was achieved after 20 min. Due to the fact that Boc-[^18^F]**BA3** was obtained in highest labelling yields with MeCN, the use of this solvent was kept for further evaluation. The second step after the radiofluorination of Boc-[^18^F]**BA3** was the cleavage of the Boc protecting group, which was performed by addition of an aqueous solution of 2 M hydrochloric acid (HCl_aq_) to the crude mixture of Boc-[^18^F]**BA3** and heating at 80 and 100 °C. At 80 °C the reaction was already completed after 5 min as detected by radio-TLC. Finally, the novel radioligand [^18^F]**BA3** could be prepared in a two-step one-pot reaction by a direct aliphatic nucleophilic substitution, starting with the precursor **9** and subsequent Boc deprotection under the optimized reaction conditions as shown in Figure 2.

#### 2.3.2. Automated Radiosynthesis

An automated radiosynthesis of [^18^F]**BA3** was established based on the optimized reaction conditions using the radiosynthesis module TRACERlab FX2 N. The detailed configuration is shown in the Appendix A. Briefly, [^18^F]fluoride (starting activities of 7–32 GBq) was trapped on a QMA cartridge and eluted into the reactor with TBAHCO_3_ dissolved in MeCN/H_2_O and dried via azeotropic distillation. The nucleophilic substitution of the bromine precursor **9** (4 mg) was performed using [^18^F]TBAF in MeCN at 100 °C for 15 min. The subsequent Boc deprotection was carried out by adding an aqueous 2 M HCl solution and stirring for 5 min at 80 °C. After neutralization with a 1 M solution of aqueous NaHCO_3_, the radiolabelled product [^18^F]**BA3** was isolated via semi-preparative RP-HPLC (Figure 2) and purified by solid-phase extraction (SPE) on a C18 cartridge. Subsequent elution with ethanol (EtOH) and formulation in sterile 0.9% saline containing 10% of EtOH reproducibly provided [^18^F]**BA3** in a total synthesis time of approximately 85 min with RCYs of 0.9 ± 0.2% (EOB, *n* = 5; see Appendix A for UV- and radio-RP-HPLC-chromatograms of formulated [^18^F]**BA3**). The identity of [^18^F]**BA3** was confirmed by analytical radio- and UV-RP-HPLC of the final product, co-injected with the reference compound **BA3** (Figure 2). Finally, the radiotracer was obtained with a radiochemical purity of >96% and a molar activity in the range of 21–51 GBq/µmol (EOS, *n* = 5).

### 2.4. Lipophilicity and Radiochemical Stability of *[^18^F]**BA3***

The distribution coefficient (logD_7.4_) was experimentally determined by the shake-flask method in the *n*-octanol/PBS system. The logD_7.4_ value of 1.75 ± 0.16 (*n* = 8) for [^18^F]**BA3** suggests that a moderate passive diffusion through the BBB is most likely [58,59,60]. The radiotracer [^18^F]**BA3** was stable in *n*-octanol, as well as in the saline formulation containing 10% EtOH at room temperature for more than two hours (radiochemical purity > 99%).

### 2.5. In Vivo PET/MRI Studies of *[^18^F]**BA3***

The pharmacokinetics of [^18^F]**BA3** was investigated by dynamic PET scans on healthy CD-1 mice in order to evaluate the passage across the BBB, nonspecific binding and thus the expected background level. The time–activity curve (TAC) of the brain (peak standardized uptake value (SUV) at 0.8 min of ~0.16) demonstrates a low brain uptake (see Figure 3 and Appendix A). Further radiotracer accumulation reaches a plateau at 15 min (~0.22 SUV) followed by a slow washout reaching a SUV of ~0.16 at 60 min after i.v. administration. Notably, only 21% of the parent compound was found in the brain at 30 min p.i., suggesting that the major part of the measured activity originates from the nonspecific accumulation of radiometabolites.

Although these first results do not support favourable imaging features of [^18^F]**BA3**, the evaluation of its potential affinity to efflux drug transporter present at the BBB could help to predict the features of other derivatives of this class of compound. Thus, pre-administration of **cyclosporine A**, an inhibitor of the P-glycoprotein (P-gp) efflux transporter, increased the initial uptake of [^18^F]**BA3** by about factor two (0.18 vs. 0.42 SUV) indicating that [^18^F]**BA3** is a substrate of P-gp.

Evaluation of the whole-body distribution derived from PET imaging revealed a low accumulation of tracer and/or metabolites in spleen, kidney, muscle, liver, and small intestines (initial uptake: SUV_5 min_ < 4), as well as a relatively constant activity in blood circulation (SUV_5 min_ to SUV_60 min_: 1.47 vs. 1.10). An elevated accumulation occurred in the gallbladder and the bladder, indicating both urinary and hepatobiliary excretion (Appendix A).

In conclusion, although [^18^F]**BA3** presents a high in vitro inhibitory potency towards HDAC1 (IC_50_ = 4.8 ± 0.6 nM) and HDAC2 (IC_50_ = 39.9 ± 3.2 nM), and an excellent specificity of HDAC1 over HDAC3 and HDAC6 (specificity ratios >230 and >2080, respectively), its low passage across the BBB and the presence of radiometabolites in brain show the need for further ligand optimization.

### 2.6. In Vivo Metabolism of *[^18^F]**BA3***

The in vivo metabolism of [^18^F]**BA3** was investigated in the plasma, brain homogenate and urine obtained from mice at 30 min after injection into the tail vein. The radiometabolites and parent compound were analysed by (i) analytical radio-RP-HPLC and (ii) radio-micellar chromatography (radio-MLC) [61]. Prior to the analysis with radio-RP-HPLC, the plasma and brain samples were treated with MeCN/H_2_O (9:1, *v*/*v*) as extraction solvent. The recovery of activity in plasma and brain samples was 97% and 95%, respectively. Radio-RP-HPLC analysis revealed that the intact radiotracer accounted for about 5% and 21% (mean, *n* = 2) of the total activity in plasma and brain, respectively. Similar results were obtained with MLC (about 6% and 24% (mean, *n* = 2) in plasma and brain, respectively), where the samples are directly injected without previous protein precipitation (for corresponding MLC chromatograms see Appendix A). In urine intact radiotracer was not detected by neither radio-RP-HPLC nor radio-MLC. In the corresponding radio-chromatograms (Figure 4) five radiometabolites ([^18^F]**M1**-[^18^F]**M5**) were detected in brain at 30 min p.i. In contrast, in plasma one single major radiometabolite, [^18^F]**M4**, amounting to 74%, was found However, a conclusion, whether the radiometabolites in the brain originate from the plasma and pass the blood–brain barrier or are formed within the brain itself, was not possible, as only a single time point was investigated.

### 2.7. Expression of HDAC1 in F98 and U251-MG Cells

The expression of HDAC1 was evaluated by immunofluorescence microscopy of the glioblastoma cell lines F98 and U251-MG to check whether they are suitable for the assessment of the toxicity of HDAC inhibitors. As shown in Figure 5, both cell lines possess a HDAC1 nucleic localization, proved by the merged signals of the HDAC1 (green) enzyme and the nuclei (blue). Interestingly, the HDAC1 signal is exactly co-localized with the nuclei in the F98 cells but additionally presents a slight cytoplasmic localization in U251-MG cells, as also previously found by Seto et al. [62].

### 2.8. The New HDAC Inhibitor ***BA3*** Has Potential to Reduce the Proliferation of Cancer Cells

Inhibition of HDAC activity has been shown to hinder the proliferation of cancer cells. To assess the cytotoxic potential of **BA3**, the effect of this compound on the proliferation of the rat glioma F98 cells and the human glioblastoma U251-MG cells in comparison with the well-established HDAC inhibitors **tacedinaline**, **vorinostat**, and **entinostat** was investigated by an MTS assay (Table 3). Although solubility problems detected only after completion of the study (slight precipitation of **BA3** at 50 µM in cell culture medium supplemented with 0.5% DMSO) preclude the direct comparison of the cytotoxic potency between the established HDAC inhibitors and **BA3**, the similar strongly diminished viability of F98 and U251-MG cells induced by **BA3** at a concentration of even lower than 50 µM indicate nevertheless a high cytotoxic potential of **BA3**. Thus, although differently designed studies would be needed to determine proper cytotoxic IC_50_ values and even though not relevant for imaging studies performed with radiotracers at low to sub-nanomolar concentrations, the presented data suggest, that the pharmacological potency of **BA3** is at least comparable if not even higher than that of established and clinically tested HDAC inhibitors.

## 3. Materials and Methods

### 3.1. General

Chemicals were purchased from commercial suppliers in analytical grade and used without further purification. Solvents were dried before use, if required. Air- and moisture-sensitive reactions were carried out under inert gas atmosphere using argon. The reaction monitoring was carried out by (radio-)TLC on pre-coated silica gel plates (Alugram^®^ Xtra SIL G/UV254; Polygram^®^ SIL G/UV254) purchased from Macherey-Nagel (Düren, Germany). The compounds were visualized with UV light at λ = 254 nm and 365 nm and/or by staining with aqueous KMnO_4_ solution or ninhydrin solution.

The UV purity of the final compounds **BA1**–**BA10** was determined by LC-MS on an UltiMate 3000 UHPLC System (Thermo Scientific, Germering, Germany) including a DAD detector (DAD-3000RS) coupled to an MSQ Plus Single Quadrupole Mass Spectrometer (Thermo Scientific, Austin, TX, USA). LC-MS analyses were performed on a ReproSil-Pur 120 C18-AQ-column (150 mm x 3 mm, 3 µm; Dr. Maisch GmbH, Ammerbuch, Germany) equipped with an appropriate precolumn in gradient mode (0–1.5 min: 5% MeCN/20 mM NH_4_OAc, 1.5–8 min: 5% → 80% MeCN/20 mM NH_4_OAc, 8–12 min: 80% MeCN/20 mM NH_4_OAc, 12–12.5 min: 80% → 5% MeCN/20 mM NH_4_OAc, 12.5–15 min: 5% MeCN/20 mM NH_4_OAc) and a flow of 0.7 mL/min. UV detection was performed at a wavelength of 254 nm and used for determination of UV purity by calculation of the relative peak area. MSQ Plus single quadrupole mass spectrometer was operated in positive electrospray ionization mode: probe temperature 500 °C, needle voltage 5 V and cone voltage 75 V.

NMR spectra (^1^H, ^13^C, ^19^F) were recorded on Mercury 300/Mercury 400 (Varian, Palo Alto, CA, USA) or Fourier 300/Avance DRX 400 Bruker (Billerica, MA, USA) instruments. The hydrogenated residue of deuterated solvents and/or tetramethylsilane (TMS) were used as internal standards for ^1^H-NMR (CDCl_3_, *δ* = 7.26; DMSO-*d_6_*, *δ* = 2.50) and ^13^C-NMR (CDCl_3_, *δ* = 77.16; DMSO-*d_6_*, *δ* = 39.52). The chemical shifts (*δ*) are reported in ppm (s, singlet; brs, broad singlet; d, doublet; t, triplet; q, quartet; m, multiplet) and the corresponding coupling constants (*J*) are reported in Hz. High resolution mass spectra (ESI +/−) were recorded on an Esquire 3000Plus instrument (Bruker Daltonics, Billerica, MA, USA; equipped with ion trap). The ^1^H-, ^13^C- and ^19^F-NMR spectra and LC-MS chromatograms for compounds **BA1**–**BA10** are displayed in Appendix A

### 3.2. General Procedures

#### 3.2.1. Suzuki Coupling **A1**

An aqueous solution of K_2_CO_3_ (2.6 eq.) was added to a solution of **1** (1 eq.) and the respective boronic acid (1.2 eq.) in 1,4-dioxane under argon atmosphere. After addition of tetrakis(triphenylphosphine)palladium(0) (0.05 eq.), the mixture was stirred under reflux until the complete consumption of **2**. The 1,4-dioxane was evaporated; the crude product was dissolved in ethyl acetate (EA) and washed with water and brine. After drying over Na_2_SO_4_ the solvent was evaporated and the product purified by flash chromatography.

#### 3.2.2. Nitro Reduction with Hydrogen **A2a**

The nitro compound was dissolved in EA and diluted with MeOH (1:1, *v*/*v*). After flushing with argon, Pd/C (0.05 eq., 10% *wt*/*wt*) was added and the mixture was stirred under hydrogen atmosphere at room temperature for 12 h. The solvent was removed, the crude product was adsorbed on silica and purified by flash chromatography.

#### 3.2.3. Nitro Reduction with Sodium Dithionite **A2b**

To a solution of nitro compound (1 eq.) in 1,4-dioxane/water (5:1, *v*/*v*), sodium dithionite (4 eq.) and sodium bicarbonate (10 eq.) were added. The heterogeneous mixture was stirred vigorously under reflux until the complete consumption of the nitro compound. The salts were filtered off and 1,4-dioxane evaporated. The organic compounds were transferred in a separation funnel with EA and H_2_O, extracted twice with EA, washed with brine, and dried over Na_2_SO_4_. After evaporation, the crude product was purified by flash chromatography.

#### 3.2.4. Amide Coupling **A3**

To a solution of the carboxylic acid and *N*,*N*-diisopropylethylamine (DIPEA) in DMF, PyBOP was added in small portions and subsequently stirred for 30 min at room temperature. In the meantime, a solution of the respective amine in DMF was prepared and then added to the stirring solution. After stirring overnight most of the DMF was evaporated, the residue was dissolved in EA, washed with water and brine, and dried over Na_2_SO_4_. After adsorbing the crude product on silica, it was purified by flash chromatography.

#### 3.2.5. Boc Cleavage **A4**

For the cleavage of the Boc group the corresponding compound (1 eq.) was dissolved in a 30% TFA (20 eq.) solution in DCM and stirred for 2–4 h. The solution was diluted with EA, washed extensively with a 5% aqueous solution of NaHCO_3_ and dried over Na_2_SO_4_. The EA was removed until the product started to precipitate. Ice-cold heptane was added, and the product was filtered off and dried under reduced pressure.

### 3.3. Compounds

#### 3.3.1. Synthesis of *tert*-butyl (4-bromo-2-nitrophenyl)carbamate (**1**)

To a solution of 4-bromo-2-nitroaniline (10 g, 46 mmol, 1 eq.) in 150 mL THF, TEA (15.9 mL, 115 mmol, 2.5 eq.) and DMAP (1.1 g, 9.2 mmol, 0.2 eq.) were added successively. While stirring at room temperature, the first portion Boc_2_O (10 g, 46 mmol, 1 eq.) was added until it was completely dissolved. Then the second portion Boc_2_O (11.6 g, 53 mmol, 1.15 eq.) was added in smaller portions and the solution was stirred at room temperature for 4 h. The reaction volume was reduced to a third and an aqueous solution of citric acid (29 g, 151 mmol) was added. After extraction of the compound with EA, the organic layer was dried over Na_2_SO_4_ and the solvent was evaporated to give a light-yellow solid. The crude product was dissolved in a solution of TFA (5.3 mL, 69 mmol, 1.5 eq.) in 140 mL DCM and stirred for 6 h. The reaction was monitored by TLC and after completion an aqueous solution of NaHCO_3_ (6.7 g, 80 mmol) was added, the product was extracted twice with DCM, washed with brine, and dried over Na_2_SO_4_. It was purified by flash chromatography (silica, gradient PE/EA 20:1 → 18:1 → 16:1) to give **1** as yellow solid (13.93 g, 95%).

^1^H-NMR (400 MHz, CDCl_3_) *δ* 9.60 (s, 1H), 8.51 (d, *J* = 9.2 Hz, 1H), 8.33 (d, *J* = 2.4 Hz, 1H), 7.68 (dd, *J* = 9.2, 2.4 Hz, 1H), 1.54 (s, 9H); ^13^C-NMR (101 MHz, CDCl_3_) *δ* 152.09, 138.67, 136.32, 135.28, 128.40, 122.35, 113.88, 82.45, 28.32 (in accordance with literature [53]).

#### 3.3.2. Synthesis of *tert*-butyl [2-nitro-4-(thiophen-2-yl)phenyl]carbamate (**2a**)

Compound **1** (2 g, 6.3 mmol) and 2-thienylboronic acid in 25 mL 1,4-dioxane, K_2_CO_3_ in 11 mL H_2_O and catalyst were reacted according to the general procedure **A1** to give after purification by flash chromatography (silica, gradient PE/EA: 25:1 → 20:1 → 15:1) compound **2a** (1.88 g, 93%) as yellow solid.

^1^H-NMR (400 MHz, CDCl_3_) *δ* 9.65 (s, 1H), 8.59 (d, *J* = 9.0 Hz, 1H), 8.40 (d, *J* = 2.3 Hz, 1H), 7.82 (dd, *J* = 8.9, 2.3 Hz, 1H), 7.37–7.30 (m, 2H), 7.10 (dd, *J* = 5.1, 3.6 Hz, 1H), 1.56 (s, 9H); ^13^C-NMR (101 MHz, CDCl_3_) *δ* 152.27, 141.49, 136.20, 134.96, 133.07, 128.88, 128.46, 125.81, 124.08, 122.52, 121.35, 82.17, 28.36 (in accordance with literature [53]).

#### 3.3.3. Synthesis of *tert*-butyl [2-nitro-4-(thiophen-3-yl)phenyl]carbamate (**2b**)

Compound **1** (3 g, 9.46 mmol) and 3-thienylboronic acid in 38 mL 1,4-dioxane, K_2_CO_3_ in 17 mL H_2_O and catalyst were reacted according to the general procedure **A1** to give after purification by flash chromatography (silica, gradient PE/EA: 24:1 → 19:1 → 14:1 → 9:1) compound **2b** (2.87 g, 95%) as yellow solid.

^1^H-NMR (400 MHz, CDCl_3_) *δ* 9.65 (s, 1H), 8.60 (d, *J* = 8.9 Hz, 1H), 8.40 (d, *J* = 2.2 Hz, 1H), 7.83 (dd, *J* = 8.9, 2.3 Hz, 1H), 7.50 (dd, *J* = 2.9, 1.4 Hz, 1H), 7.45–7.35 (m, 2H), 1.56 (s, 9H); ^13^C-NMR (101 MHz, CDCl_3_) *δ* 152.33, 139.55, 136.25, 134.78, 133.62, 130.16, 127.16, 125.90, 123.08, 121.27, 121.21, 82.07, 28.36.

#### 3.3.4. Synthesis of *tert*-butyl (4′-fluoro-3-nitro-[1,1′-biphenyl]-4-yl)carbamate (**2c**)

Compound **1** (1 g, 3.15 mmol) and 4-fluorophenylboronic acid in 15 mL 1,4-dioxane, K_2_CO_3_ in 5 mL H_2_O and catalyst were reacted according to the general procedure **A1** to give after purification by flash chromatography (silica, gradient PE/EA: 25:1 → 11.5:1 → 4:1) compound **2c** (1.0 g, 96%) as white solid.

^1^H-NMR (400 MHz, CDCl_3_) *δ* 9.66 (s, 1H), 8.63 (d, *J* = 8.9 Hz, 1H), 8.37 (d, *J* = 2.3 Hz, 1H), 7.79 (dd, *J* = 8.9, 2.3 Hz, 1H), 7.59–7.50 (m, 2H), 7.21–7.10 (m, 2H), 1.56 (s, 9H); ^19^F-NMR (377 MHz, CDCl_3_) *δ* −114.12 (ddd, *J* = 13.8, 8.6, 5.2 Hz); ^13^C-NMR (76 MHz, CDCl_3_) *δ* 162.98 (d, *J* = 247.9 Hz), 152.35, 136.34, 135.08, 134.51 (d, *J* = 3.2 Hz), 134.28, 134.12, 128.55 (d, *J* = 8.2 Hz), 123.73, 121.41, 116.21 (d, *J* = 21.7 Hz), 82.16, 28.36 (in accordance with literature [55]).

#### 3.3.5. Synthesis of *tert*-butyl [4-(furan-2-yl)-2-nitrophenyl]carbamate (**2d**)

Compound **1** (2 g, 6.3 mmol) and 2-furanylboronic acid in 25 mL 1,4-dioxane, K_2_CO_3_ in 11 mL H_2_O and catalyst were reacted according to the general procedure **A1** to give after purification by flash chromatography (silica, gradient PE/EA: 30:1 → 20:1 → 10:1) compound **2d** (1.83 g, 95%) as orange solid.

^1^H-NMR (400 MHz, CDCl_3_) *δ* 9.68 (s, 1H), 8.62 (d, *J* = 8.9 Hz, 1H), 8.48 (d, *J* = 2.1 Hz, 1H), 7.88 (dd, *J* = 8.9, 2.2 Hz, 1H), 7.52 (dd, *J* = 1.8, 0.7 Hz, 1H), 6.72 (dd, *J* = 3.4, 0.8 Hz, 1H), 6.52 (dd, *J* = 3.4, 1.8 Hz, 1H), 1.58 (s, 9H); ^13^C-NMR (101 MHz, CDCl_3_) *δ* 152.28, 151.53, 142.87, 136.25, 134.77, 130.92, 125.37, 121.19, 120.66, 112.08, 106.12, 82.12, 28.36.

#### 3.3.6. Synthesis of *tert*-butyl [4-(furan-3-yl)-2-nitrophenyl]carbamate (**2e**)

Compound **1** (2 g, 6.3 mmol) and 3-furanylboronic acid in 25 mL 1,4-dioxane, K_2_CO_3_ in 11 mL H_2_O and catalyst were reacted according to the general procedure **A1** to give after purification by flash chromatography (silica, gradient PE/EA: 30:1 → 20:1 → 10:1) compound **2d** (1.73 g, 90%) as orange solid.

^1^H-NMR (400 MHz, CDCl_3_) *δ* 9.63 (s, 1H), 8.57 (d, *J* = 8.9 Hz, 1H), 8.27 (d, *J* = 2.2 Hz, 1H), 7.77 (t, *J* = 1.2 Hz, 1H), 7.70 (dd, *J* = 8.9, 2.2 Hz, 1H), 7.53–7.48 (m, 1H), 6.70 (dd, *J* = 2.0, 0.9 Hz, 1H), 1.55 (s, 9H); ^13^C-NMR (101 MHz, CDCl_3_) *δ* 152.35, 144.35, 139.07, 136.30, 134.70, 133.17, 126.96, 124.39, 122.54, 121.34, 108.61, 82.09, 28.37.

#### 3.3.7. Synthesis of *tert*-butyl [2-amino-4-(thiophen-2-yl)phenyl]carbamate (**3a**)

The reduction in compound **2a** (1.5 g, 4.7 mmol) was performed according to the general procedure **A2a** in 60 mL EA/MeOH (1:1, *v*/*v*) to give after purification by flash chromatography (silica, gradient PE/EA: 6:1 → 4:1 → 3:1 → 1:1) compound **3a** (1.37 g) as white-beige solid in quantitative yield.

^1^H-NMR (400 MHz, DMSO-*d_6_*) *δ* 8.34 (s, 1H), 7.44 (dd, *J* = 5.1, 1.2 Hz, 1H), 7.32–7.24 (m, 2H), 7.08 (dd, *J* = 5.1, 3.6 Hz, 1H), 6.98 (d, *J* = 2.2 Hz, 1H), 6.84 (dd, *J* = 8.2, 2.2 Hz, 1H), 5.01 (s, 2H), 1.47 (s, 9H); ^13^C-NMR (101 MHz, DMSO-*d_6_*) *δ* 153.48, 144.01, 141.20, 130.18, 128.21, 124.55, 123.51, 122.42, 113.65, 112.40, 78.83, 28.15 (in accordance with literature [49]).

#### 3.3.8. Synthesis of *tert*-butyl [2-amino-4-(thiophen-3-yl)phenyl]carbamate (**3b**)

The reduction in compound **2b** (1.0 g, 3.12 mmol) was performed according to the general procedure **A2a** in 50 mL EA/MeOH (1:1, *v*/*v*) to give after purification by flash chromatography (silica, gradient PE/EA: 6:1 → 4:1 → 3:1 → 1:1) compound **3b** (890 mg, 98%) as white-beige solid.

^1^H-NMR (400 MHz, DMSO-*d_6_*) *δ* 8.32 (s, 1H), 7.66–7.54 (m, 2H), 7.38 (dd, *J* = 5.0, 1.4 Hz, 1H), 7.25 (d, *J* = 8.2 Hz, 1H), 7.01 (d, *J* = 2.1 Hz, 1H), 6.88 (dd, *J* = 8.2, 2.1 Hz, 1H), 4.90 (s, 2H), 1.47 (s, 9H); ^13^C-NMR (101 MHz, DMSO-*d_6_*) *δ* 153.54, 141.82, 141.12, 131.75, 126.70, 125.99, 124.47, 123.09, 119.54, 114.46, 113.18, 78.73, 28.16.

#### 3.3.9. Synthesis of *tert*-butyl (3-amino-4′-fluoro-[1,1′-biphenyl]-4-yl)carbamate (**3c**)

The reduction in compound **2c** (1.0 g, 3.0 mmol) was performed according to the general procedure **A2a** in 50 mL EA/MeOH (1:1, *v*/*v*) to give after purification by flash chromatography (silica, gradient PE/EA: 4:1 → 1.5:1 → 1:1) compound **3c** (895 mg) as white-beige solid in quantitative yield.

^1^H-NMR (300 MHz, CDCl_3_) *δ* 7.52–7.43 (m, 2H), 7.34 (d, *J* = 7.8 Hz, 1H), 7.13–7.03 (m, 2H), 6.99–6.90 (m, 2H), 6.24 (s, 1H), 3.85 (s, 2H), 1.53 (s, 9H); ^19^F-NMR (282 MHz, CDCl_3_) *δ* −116.04 (^1^H decoupled); ^13^C-NMR (75 MHz, CDCl_3_) *δ* 162.94 (d, *J* = 246.1 Hz), 153.96, 140.23, 138.41, 137.15, 128.59 (d, *J* = 8.02 Hz), 125.11, 124.31, 118.56, 116.25, 115.63 (d, *J* = 21.38 Hz), 80.86, 28.48 (in accordance with literature [55]).

#### 3.3.10. Synthesis of *tert*-butyl [2-amino-4-(furan-2-yl)phenyl]carbamate (**3d**)

The reduction in compound **2d** (1.78 g, 5.85 mmol) was performed according to the general procedure **A2b** in 42 mL 1,4-dioxane/H_2_O (5:1, *v*/*v*) to give after purification by flash chromatography (silica, gradient PE/EA: 4:1 → 2:1 → 1:1) compound **3d** (1.34 g, 83%) as white-beige solid.

^1^H-NMR (400 MHz, CDCl_3_) *δ* 7.42 (dd, *J* = 1.8, 0.8 Hz, 1H), 7.32 (d, *J* = 8.3 Hz, 1H), 7.14–7.07 (m, 2H), 6.55 (dd, *J* = 3.4, 0.8 Hz, 1H), 6.44 (dd, *J* = 3.4, 1.8 Hz, 1H), 6.26 (s, 1H), 3.77 (s, 2H), 1.52 (s, 9H); ^13^C-NMR (101 MHz, CDCl_3_) *δ* 153.90, 153.86, 141.88, 139.86, 128.80, 124.70, 124.46, 115.71, 113.03, 111.71, 104.73, 80.80, 28.47 (in accordance with literature [63]).

#### 3.3.11. Synthesis of *tert*-butyl [2-amino-4-(furan-3-yl)phenyl]carbamate (**3e**)

The reduction in compound **2e** (1.68 g, 5.52 mmol) was performed according to the general procedure **A2b** in 42 mL 1,4-dioxane/H_2_O (5:1, *v*/*v*) to give after purification by flash chromatography (silica, gradient PE/EA: 4:1 → 2:1 → 1:1) compound **3e** (611 mg, 40%) as white-beige solid.

^1^H-NMR (400 MHz, CDCl_3_) *δ* 7.65 (dd, *J* = 1.6, 0.9 Hz, 1H), 7.45–7.43 (m, 1H), 7.27 (d, *J* = 8.0 Hz, 1H), 6.95–6.86 (m, 2H), 6.63 (dd, *J* = 1.9, 0.9 Hz, 1H), 6.23 (s, 1H), 3.42 (brs, 2H), 1.52 (s, 9H); ^13^C-NMR (101 MHz, CDCl_3_) *δ* 153.98, 143.63, 140.27, 138.45, 130.55, 126.25, 125.21, 123.91, 117.52, 115.06, 109.04, 80.77, 28.47.

#### 3.3.12. Synthesis of *tert*-butyl (2-{4-[(2-fluoropropanamido)methyl]benzamido}-4-[thiophen-2-yl]phenyl)carbamate (**4a**)

Compound **7** (20 mg, 0.089 mmol, 1 eq.), compound **3a** (29 mg, 0.098 mmol, 1.1 eq.), DIPEA (34 µL, 0.2 mmol, 2.3 eq.) and PyBOP (56 mg, 0.107 mmol, 1.2 eq.) in 1 mL DMF were reacted according to the general procedure **A3** to give after purification by flash chromatography (silica, gradient PE/EA: 3:1 → 2:1) compound **4a** (40 mg, 90%) as brownish foam.

^1^H-NMR (400 MHz, CDCl_3_) *δ* 9.35 (s, 1H), 7.96–7.87 (m, 3H), 7.35–7.25 (m, 4H), 7.24–7.19 (m, 2H), 7.16 (s, 1H), 7.05–6.98 (m, 1H), 6.82 (d, *J* = 5.5 Hz, 1H), 5.06 (dq, *J* = 49.3, 6.8 Hz, 1H), 4.51 (d, *J* = 6.0 Hz, 2H), 1.61 (dd, *J* = 24.8, 6.8 Hz, 3H), 1.48 (s, 9H); ^19^F-NMR (377 MHz, DMSO-*d_6_*) *δ* −181.13 (dqd, *J* = 51.5, 25.6, 19.3 Hz); ^13^C-NMR (101 MHz, CDCl_3_) *δ* 171.00 (d, *J* = 19.2 Hz), 165.68, 154.69, 143.34, 141.89, 133.40, 132.10, 130.88, 129.64, 128.10, 128.07, 127.76, 124.96, 123.55, 123.45, 122.98, 88.98 (d, *J* = 182.9 Hz), 81.44, 42.62, 28.38, 18.65 (d, *J* = 21.5 Hz); HRFT-MS (ESI+): *m*/*z* = 995.3654 (calc. 995.3642 for [2M+H]^+^).

#### 3.3.13. Synthesis of *tert*-butyl (2-{4-[(2-fluoropropanamido)methyl]benzamido}-4-[thiophen-3-yl]phenyl)carbamate (**4b**)

Compound **7** (12 mg, 0.053 mmol, 1 eq.), compound **3b** (17 mg, 0.058 mmol, 1.1 eq.), DIPEA (21 µL, 0.122 mmol, 2.3 eq.) and PyBOP (34 mg, 0.064 mmol, 1.2 eq.) in 1 mL DMF were reacted according to the general procedure **A3** to give after a rough purification by flash chromatography (silica, gradient PE/EA: 3:1 → 2:1) compound **4b** (24 mg, 91%) as brownish foam.

^1^H-NMR (400 MHz, CDCl_3_) *δ* 9.34 (s, 1H), 7.95–7.86 (m, 3H), 7.36–7.22 (m, 7H), 7.08 (s, 1H), 6.79 (d, *J* = 5.4 Hz, 1H), 5.05 (dq, *J* = 49.3, 6.8 Hz, 1H), 4.50 (d, *J* = 6.0 Hz, 2H), 1.60 (dd, *J* = 24.8, 6.8 Hz, 3H), 1.47 (s, 9H); ^19^F-NMR (377 MHz, DMSO-*d_6_*) *δ* −181.16 (tdd, *J* = 53.1, 44.7, 25.2 Hz); ^13^C-NMR (101 MHz, CDCl_3_) *δ* 170.98 (d, *J* = 19.3 Hz), 165.60, 154.77, 141.89, 141.27, 133.62, 133.49, 130.97, 129.22, 128.05, 127.79, 126.36, 126.27, 124.95, 123.99, 123.62, 120.53, 88.99 (d, *J* = 182.8 Hz), 81.43, 42.63, 28.40, 18.65 (d, *J* = 21.5 Hz); HRFT-MS (ESI+): *m*/*z* = 1017.3456 (calc. 1017.3461 for [2M+Na]^+^).

#### 3.3.14. Synthesis of *tert*-butyl (4′-fluoro-3-{4-[(2-fluoropropanamido)methyl]benzamido}-[1,1′-biphenyl]-4-yl)carbamate (**4c**)

Compound **7** (100 mg, 0.444 mmol, 1 eq.), compound **3c** (174 mg, 0.577 mmol, 1.3 eq.), DIPEA (184 µL, 1.021 mmol, 2.3 eq.) and PyBOP (277 mg, 0.533 mmol, 1.2 eq.) in 12 mL DMF were reacted according to the general procedure **A3** to give after a rough purification by flash chromatography (silica, gradient PE/EA: 9:1 → 4:1 → 1:1) the crude compound **4c** (300 mg, 132% mass balance). It was used without further purification for following reactions.

^1^H-NMR (300 MHz, DMSO-*d_6_*) *δ* 9.89 (s, 1H), 8.79 (d, *J* = 9.7 Hz, 2H), 7.98–7.89 (m, 2H), 7.84 (d, *J* = 2.2 Hz, 1H), 7.76–7.60 (m, 3H), 7.50 (dd, *J* = 8.5, 2.2 Hz, 1H), 7.42 (d, *J* = 8.3 Hz, 2H), 7.34–7.25 (m, 2H), 5.09 (dq, *J* = 48.9, 6.7 Hz, 1H), 4.40 (d, *J* = 6.1 Hz, 2H), 1.47 (dd, *J* = 24.5, 6.7 Hz, 3H), 1.47 (s, 9H); ^19^F-NMR (282 MHz, DMSO-*d_6_*) *δ* −115.65 (ddd, *J* = 14.1, 9.0, 5.3 Hz), −181.12 (dd, *J* = 48.9, 24.3 Hz); ^13^C-NMR (75 MHz, DMSO-*d_6_*) *δ* 170.02 (d, *J* = 20.2 Hz), 165.29, 161.78 (d, *J* = 244.3 Hz), 153.41, 143.34, 135.81 (d, *J* = 3.0 Hz), 134.82, 132.79, 131.09, 129.95, 128.34 (d, *J* = 8.3 Hz), 127.69, 127.11, 124.07, 124.03, 123.72, 115.75 (d, *J* = 21.3 Hz), 87.96 (d, *J* = 179.2 Hz), 79.79, 41.49, 28.03, 18.46 (d, *J* = 21.9 Hz); HRFT-MS (ESI+): *m*/*z* = 1019.4312 (calc. 1019.4325 for [2M+H]^+^).

#### 3.3.15. Synthesis of *tert*-butyl (2-{4-[(2-fluoropropanamido)methyl]benzamido}-4-[furan-2-yl]phenyl)carbamate (**4d**)

Compound **7** (130 mg, 0.577 mmol, 1 eq.), compound **3d** (206 mg, 0.75 mmol, 1.3 eq.), DIPEA (238 µL, 1.327 mmol, 2.3 eq.) and PyBOP (360 mg, 0.692 mmol, 1.2 eq.) in 15 mL DMF were reacted according to the general procedure **A3** to give after a rough purification by flash chromatography (silica, gradient PE/EA: 10:1 → 8:1 → 4:1 → 1:1) the crude compound **4d** (290 mg, 104%). It was used without further purification for following reactions.

^1^H-NMR (400 MHz, DMSO-*d_6_*) *δ* 9.87 (s, 1H), 8.84–8.78 (m, 1H), 8.75 (s, 1H), 7.94 (d, *J* = 8.3 Hz, 2H), 7.88 (d, *J* = 2.0 Hz, 1H), 7.73 (dd, *J* = 1.8, 0.8 Hz, 1H), 7.62 (d, *J* = 8.6 Hz, 1H), 7.54 (dd, *J* = 8.5, 2.1 Hz, 1H), 7.42 (d, *J* = 8.3 Hz, 2H), 6.88 (dd, *J* = 3.4, 0.8 Hz, 1H), 6.59 (dd, *J* = 3.4, 1.8 Hz, 1H), 5.09 (dq, *J* = 48.9, 6.7 Hz, 1H), 4.40 (d, *J* = 6.2 Hz, 2H), 1.47 (dd, *J* = 24.5, 6.7 Hz, 3H), 1.46 (s, 9H); ^19^F-NMR (377 MHz, DMSO-*d_6_*) *δ* −176.35 (dq, *J* = 49.0, 24.6 Hz); ^13^C-NMR (101 MHz, DMSO-*d_6_*) *δ* 170.03 (d, *J* = 20.3 Hz), 165.32, 153.34, 152.47, 143.34, 142.74, 132.76, 131.05, 129.89, 127.73, 127.11, 126.27, 123.98, 120.99, 120.86, 112.10, 105.46, 87.97 (d, *J* = 179.2 Hz), 79.80, 41.50, 28.03, 18.47 (d, *J* = 21.9 Hz); HRFT-MS (ESI+): *m*/*z* = 963.4101 (calc. 963.4099 for [2M+H]^+^).

#### 3.3.16. Synthesis of *tert*-butyl (2-{4-[(2-fluoropropanamido)methyl]benzamido}-4-[furan-3-yl]phenyl)carbamate (**4e**)

Compound **7** (130 mg, 0.577 mmol, 1 eq.), compound **3e** (206 mg, 0.75 mmol, 1.3 eq.), DIPEA (238 µL, 1.327 mmol, 2.3 eq.) and PyBOP (360 mg, 0.692 mmol, 1.2 eq.) in 15 mL DMF were reacted according to the general procedure **A3** to give after a rough purification by flash chromatography (silica, gradient PE/EA: 10:1 → 8:1 → 4:1 → 1:1) the crude compound **4e** (265 mg, 95%). It was used without further purification for following reactions.

^1^H-NMR (400 MHz, DMSO-*d_6_*) *δ* 9.86 (s, 1H), 8.84–8.74 (m, 1H), 8.65 (s, 1H), 8.14 (dd, *J* = 1.6, 0.9 Hz, 1H), 8.02–7.86 (m, 2H), 7.77–7.69 (m, 2H), 7.60–7.50 (m, 1H), 7.49–7.33 (m, 3H), 6.91 (dd, *J* = 1.9, 0.9 Hz, 1H), 5.09 (ddd, *J* = 48.8, 6.7, 5.0 Hz, 1H), 4.43–4.34 (m, 2H), 1.52–1.40 (m, 12H); ^19^F-NMR (377 MHz, DMSO-*d_6_*) *δ* −181.13 (d, *J* = 17.6 Hz); ^13^C-NMR (101 MHz, DMSO-*d_6_*) *δ* 170.03 (d, *J* = 20.2 Hz), 165.24, 153.39, 144.31, 143.32, 139.10, 132.74, 130.77, 129.92, 129.38, 127.69, 127.10, 125.18, 123.95, 123.23, 122.99, 108.64, 87.96 (d, *J* = 179.4 Hz), 79.69, 41.50, 28.03, 18.46 (d, *J* = 21.7 Hz); HRFT-MS (ESI+): *m*/*z* = 504.1935 (calc. 504.1905 for [M+Na]^+^).

#### 3.3.17. Synthesis of *tert*-butyl {2-[4-(2-fluoropropanamido)benzamido]-4-(thiophen-2-yl)phenyl}carbamate (**5a**)

Compound **8** (300 mg, 1.42 mmol, 1 eq.), compound **3a** (578 mg, 1.988 mmol, 1.4 eq.), DIPEA (612 µL, 3.408 mmol, 2.4 eq.) and PyBOP (887 mg, 1.704 mmol, 1.2 eq.) in 15 mL DMF were reacted according to the general procedure **A3** to give after purification by flash chromatography (silica, gradient PE/EA: 5:1 → 4:1 → 3:1) the compound **5a** (500 mg, 73%) as brownish foam.

^1^H-NMR (400 MHz, DMSO-*d_6_*) *δ* 10.38 (d, *J* = 1.7 Hz, 1H), 9.87 (s, 1H), 8.74 (s, 1H), 8.02–7.94 (m, 2H), 7.91–7.85 (m, 2H), 7.84 (d, *J* = 2.2 Hz, 1H), 7.61 (d, *J* = 8.5 Hz, 1H), 7.54–7.49 (m, 2H), 7.46 (dd, *J* = 3.6, 1.2 Hz, 1H), 7.13 (dd, *J* = 5.1, 3.6 Hz, 1H), 5.23 (dq, *J* = 48.5, 6.7 Hz, 1H), 1.56 (dd, *J* = 24.5, 6.7 Hz, 3H), 1.46 (s, 9H); ^19^F-NMR (377 MHz, DMSO-*d_6_*) *δ* −179.83 (^1^H decoupled); ^13^C-NMR (101 MHz, DMSO-*d_6_*) *δ* 168.92 (d, *J* = 20.7 Hz), 164.91, 153.33, 142.68, 141.42, 131.17, 130.07, 129.64, 129.19, 128.56, 128.49, 125.42, 124.15, 123.42, 122.82, 122.59, 119.30, 87.78 (d, *J* = 179.1 Hz), 79.81, 28.02, 18.25 (d, *J* = 22.1 Hz); HRFT-MS (ESI+): *m*/*z* = 967.3345 (calc. 967.3329 for [2M+H]^+^).

#### 3.3.18. Synthesis of *tert*-butyl {2-[4-(2-fluoropropanamido)benzamido]-4-(thiophen-3-yl)phenyl}carbamate (**5b**)

Compound **8** (164 mg, 0.774 mmol, 1.5 eq.), compound **3b** (150 mg, 0.516 mmol, 1 eq.), DIPEA (360 µL, 2.064 mmol, 4 eq.) and PyBOP (540 mg, 1.032 mmol, 2 eq.) in 10 mL DMF were reacted according to the general procedure **A3** to give after purification by flash chromatography (silica, gradient PE/EA: 5:1 → 4:1 → 3:1) the compound **5b** (220 mg, 88%) as brownish foam.

^1^H-NMR (400 MHz, DMSO-*d_6_*) *δ* 10.38 (d, *J* = 1.7 Hz, 1H), 9.85 (s, 1H), 8.69 (s, 1H), 8.01–7.94 (m, 2H), 7.89–7.84 (m, 3H), 7.81 (dd, *J* = 2.9, 1.4 Hz, 1H), 7.64 (dd, *J* = 5.0, 2.9 Hz, 1H), 7.60 (d, *J* = 8.5 Hz, 1H), 7.56 (dd, *J* = 8.5, 2.0 Hz, 1H), 7.52 (dd, *J* = 5.0, 1.4 Hz, 1H), 5.23 (dq, *J* = 48.5, 6.7 Hz, 1H), 1.55 (dd, *J* = 24.6, 6.7 Hz, 3H), 1.46 (s, 9H); ^19^F-NMR (377 MHz, DMSO-*d_6_*) *δ* −179.84 (^1^H decoupled); ^13^C-NMR (101 MHz, DMSO-*d_6_*) *δ* 168.91 (d, *J* = 20.7 Hz), 164.83, 153.38, 141.37, 140.66, 131.10, 130.88, 129.94, 129.24, 128.50, 127.14, 126.01, 123.93, 123.70, 123.37, 120.56, 119.30, 87.77 (d, *J* = 179.1 Hz), 79.73, 28.03, 18.25 (d, *J* = 22.2 Hz); HRFT-MS (ESI+): *m*/*z* = 989.3136 (calc. 989.3148 for [2M+Na]^+^).

#### 3.3.19. Synthesis of *tert*-butyl {4′-fluoro-3-[4-(2-fluoropropanamido)benzamido]-[1,1′-biphenyl]-4-yl}carbamate (**5c**)

Compound **8** (100 mg, 0.47 mmol, 1 eq.), compound **3c** (187 mg, 0.616 mmol, 1.3 eq.), DIPEA (196 µL, 1.09 mmol, 2.3 eq.) and PyBOP (296 mg, 0.569 mmol, 1.2 eq.) in 12 mL DMF were reacted according to the general procedure **A3** to give after purification by flash chromatography (silica, gradient PE/EA: 8:1 → 7:1 → 6:1 → 4:1 → 2:1 → 1:2) the compound **5c** (185 mg, 79%) as brownish foam.

^1^H-NMR (400 MHz, DMSO-*d_6_*) *δ* 10.38 (s, 1H), 9.86 (s, 1H), 8.78 (s, 1H), 8.01–7.94 (m, 2H), 7.88–7.77 (m, 3H), 7.73–7.60 (m, 3H), 7.49 (dd, *J* = 8.5, 2.2 Hz, 1H), 7.33–7.24 (m, 2H), 5.23 (dq, *J* = 48.5, 6.7 Hz, 1H), 1.55 (dd, *J* = 24.6, 6.7 Hz, 3H), 1.47 (s, 9H); ^19^F-NMR (377 MHz, DMSO-*d_6_*) *δ* −115.63 (tt, *J* = 9.9, 5.4 Hz), −179.85 (dq, *J* = 49.1, 24.4 Hz); ^13^C-NMR (101 MHz, DMSO-*d_6_*) *δ* 168.92 (d, *J* = 20.5 Hz), 164.85, 161.78 (d, *J* = 244.2 Hz), 153.43, 141.40, 135.83, 134.83, 131.10, 130.26, 130.03, 129.26, 128.50, 128.34 (d, *J* = 8.1 Hz), 124.08, 123.68, 119.32, 115.75 (d, *J* = 21.3 Hz), 87.77 (d, *J* = 179.1 Hz), 79.81, 28.03, 18.25 (d, *J* = 22.1 Hz); HRFT-MS (ESI+): *m*/*z* = 991.3996 (calc. 991.4012 for [2M+H]^+^).

#### 3.3.20. Synthesis of *tert*-butyl {2-[4-(2-fluoropropanamido)benzamido]-4-(furan-2-yl)phenyl}carbamate (**5d**)

Compound **8** (130 mg, 0.616 mmol, 1 eq.), compound **3d** (220 mg, 0.801 mmol, 1.3 eq.), DIPEA (255 µL, 1.417 mmol, 2.3 eq.) and PyBOP (385 mg, 0.739 mmol, 1.2 eq.) in 15 mL DMF were reacted according to the general procedure **A3** to give after purification by flash chromatography (silica, gradient PE/EA: 12:1 → 10:1 → 8:1 → 6:1 → 4:1) the compound **5d** (260 mg, 90%) as brownish foam.

^1^H-NMR (400 MHz, DMSO-*d_6_*) *δ* 10.38 (s, 1H), 9.84 (s, 1H), 8.75 (s, 1H), 8.01–7.93 (m, 2H), 7.90–7.82 (m, 3H), 7.75–7.70 (m, 1H), 7.62 (d, *J* = 8.5 Hz, 1H), 7.54 (dd, *J* = 8.5, 2.1 Hz, 1H), 6.88 (dd, *J* = 3.4, 0.8 Hz, 1H), 6.58 (dd, *J* = 3.4, 1.8 Hz, 1H), 5.23 (dq, *J* = 48.5, 6.7 Hz, 1H), 1.56 (dd, *J* = 24.5, 6.7 Hz, 3H), 1.46 (s, 9H); ^19^F-NMR (377 MHz, DMSO-*d_6_*) *δ* −179.85 (dq, *J* = 48.9, 24.6 Hz); ^13^C-NMR (101 MHz, DMSO-*d_6_*) *δ* 168.92 (d, *J* = 20.6 Hz), 164.88, 153.36, 152.48, 142.73, 141.40, 131.06, 129.97, 129.24, 128.53, 126.29, 124.00, 121.03, 120.80, 119.32, 112.10, 105.45, 87.78 (d, *J* = 179.2 Hz), 79.81, 28.03, 18.26 (d, *J* = 22.1 Hz); HRFT-MS (ESI+): *m*/*z* = 935.3795 (calc. 935.3786 for [2M+H]^+^).

#### 3.3.21. Synthesis of *tert*-butyl {2-[4-(2-fluoropropanamido)benzamido]-4-(furan-3-yl)phenyl}carbamate (**5e**)

Compound **8** (177 mg, 0.84 mmol, 1 eq.), compound **3e** (300 mg, 1.09 mmol, 1.3 eq.), DIPEA (347 µL, 1.93 mmol, 2.3 eq.) and PyBOP (525 mg, 1 mmol, 1.2 eq.) in 20 mL DMF were reacted according to the general procedure **A3** to give after purification by flash chromatography (silica, gradient PE/EA: 8:2 → 7:3 → 6:4) the compound **5e** (224 mg, 57%) as brownish foam.

^1^H-NMR (300 MHz, DMSO-*d_6_*) *δ* 10.37 (s, 1H), 9.84 (s, 1H), 8.65 (s, 1H), 8.16–8.14 (m, 1H), 8.02–7.94 (m, 2H), 7.88–7.84 (m, 2H), 7.75–7.72 (m, 2H), 7.57 (d, *J* = 8.4 Hz, 1H), 7.46 (dd, *J* = 8.4, 2.1 Hz, 1H), 6.92 (dd, *J* = 1.9, 0.9 Hz, 1H), 5.23 (dq, *J* = 48.5, 6.7 Hz, 1H), 1.56 (dd, *J* = 24.5, 6.7 Hz, 3H), 1.46 (s, 9H); ^19^F-NMR (282 MHz, DMSO-*d_6_*) *δ* −179.85 (dq, *J* = 48.9, 24.5 Hz); ^13^C-NMR (75 MHz, DMSO-*d_6_*) *δ* 169.38 (d, *J* = 20.7 Hz), 165.27, 153.85, 144.77, 141.84, 139.57, 131.25, 130.44, 129.68, 128.96, 128.34, 125.65, 124.40, 123.75, 123.41, 119.76, 109.10, 88.24 (d, *J* = 179.2 Hz), 80.16, 28.50, 18.71 (d, *J* = 22.1 Hz); HRFT-MS (ESI+): *m*/*z* = 368.1418 (calc. 368.1405 for [M+H]^+^ Boc deprotection).

#### 3.3.22. Synthesis of 2-fluoropropanoyl chloride (**6**)

Sodium 2-fluoropropanoate (3 g, 26.3 mmol, 1 eq.) was poured in a two-necked flask, cooled down to 0 °C in an ice bath and stirred vigorously. Phosphorus pentachloride PCl_5_ (6 g, 28.9 mmol, 1.1 eq.) was then added in small portions. After the complete addition of PCl_5_ the mixture was stirred further at room temperature for 3 h. After distillation (10 cm column, T_bath_ = 95–105 °C, T_B_ = 75–80 °C; T_B,Lit_ = 80.5–81 °C [64]) compound **6** (1.7 g, 58%) was isolated as a clear liquid.

Refractive index n_D_^20^ = 1.384 (n_D_^20^_Lit_ = 1.384 [64]); ^1^H-NMR (400 MHz, CDCl_3_) *δ* 5.14 (dq, *J* = 48.6, 6.9 Hz, 1H), 1.70 (dd, *J* = 22.8, 6.9 Hz, 3H); ^13^C-NMR (101 MHz, CDCl_3_) *δ* 172.57 (d, *J* = 27.5 Hz), 90.33 (d, *J* = 194.0 Hz), 17.70 (d, *J* = 22.2 Hz) (according to literature [65]).

#### 3.3.23. Synthesis of 4-[(2-fluoropropanamido)methyl]benzoic acid (**7**)

To an ice-cold suspension of **6** (1 g, 9 mmol, 1.2 eq.) and 4-(aminomethyl)benzoic acid (1.14 g, 7.5 mmol, 1 eq.) in 13 mL chloroform, pyridine (730 µL, 9 mmol, 1.2 eq.) was added dropwise. After stirring at room temperature overnight, most of the chloroform was evaporated and the residue dissolved in an aqueous solution of Na_2_CO_3_ (2.12 g, 20 mmol in 20 mL H_2_O). After washing twice with MTBE, the aqueous solution was acidified with 10 mL of a 6 M HCl solution, and the product started to precipitate. After stirring for a while, the solid was filtered off and dried under reduced pressure to give compound **7** (1.06 g, 63%) as white solid.

^1^H-NMR (400 MHz, DMSO-*d_6_*) *δ* 12.86 (s, 1H), 8.82–8.74 (m, 1H), 7.92–7.88 (m, 2H), 7.38–7.34 (m, 2H), 5.08 (dq, *J* = 48.8, 6.7 Hz, 1H), 4.37 (d, *J* = 6.1 Hz, 2H), 1.46 (dd, *J* = 24.5, 6.7 Hz, 3H); ^19^F-NMR (377 MHz, DMSO-*d_6_*) *δ* −181.15 (^1^H decoupled); ^13^C-NMR (101 MHz, DMSO-*d_6_*) *δ* 170.04 (d, *J* = 20.2 Hz), 167.16, 144.32, 129.39, 129.34, 127.14, 87.96 (d, *J* = 179.2 Hz), 41.53, 18.46 (d, *J* = 21.9 Hz).

#### 3.3.24. Synthesis of 4-(2-fluoropropanamido)benzoic acid (**8**)

To an ice-cold suspension of **6** (1 g, 9 mmol, 1.2 eq.) and 4-aminobenzoic acid (1.03 g, 7.5 mmol, 1.2 eq.) in 13 mL chloroform, pyridine (730 µL, 9 mmol, 1.2 eq.) was added dropwise. After stirring at room temperature overnight, most of the chloroform was evaporated and the residue dissolved in an aqueous solution of Na_2_CO_3_ (2.12 g, 20 mmol in 20 mL H_2_O). After washing twice with MTBE, the aqueous solution was acidified with 10 mL of a 6 M HCl solution and the product started to precipitate. After stirring for a while, the solid was filtered off and dried under reduced pressure to give compound **8** (1.25 g, 79%) as white solid.

^1^H-NMR (400 MHz, DMSO-*d_6_*) *δ* 12.74 (s, 1H), 10.36 (s, 1H), 7.97–7.85 (m, 2H), 7.84–7.76 (m, 2H), 5.21 (dq, *J* = 48.5, 6.7 Hz, 1H), 1.53 (dd, *J* = 24.5, 6.7 Hz, 3H); ^19^F-NMR (377 MHz, DMSO-*d_6_*) *δ* −179.89 (^1^H decoupled); ^13^C-NMR (101 MHz, DMSO-*d_6_*) *δ* 168.95 (d, *J* = 20.7 Hz), 166.86, 142.20, 130.27, 125.88, 119.30, 87.73 (d, *J* = 179.0 Hz), 18.27 (d, *J* = 22.2 Hz).

#### 3.3.25. Synthesis of *tert*-butyl (2-{4-[(2-bromopropanamido)methyl]benzamido}-4-[thiophen-3-yl]phenyl)carbamate (**9**)

To a solution of 4-(Fmoc-aminomethyl)benzoic acid (350 mg, 0.94 mmol, 1 eq.) and DIPEA (337 µL, 1.87 mmol, 2 eq.) in 7 mL DMF, PyBOP (536 mg, 1.03 mmol, 1.1 eq.) was added in small portions. After stirring the mixture for 30 min, a solution of **3b** (355 mg, 1.22 mmol, 1.3 eq.) in 3 mL DMF was added and the solution was stirred at r.t. overnight. The amount of the DMF was reduced by evaporation, the residue was dissolved in EA, washed with water and brine and the organic phase was dried over Na_2_SO_4_. A filtration over a silica pad (silica, gradient PE/EA: 10:0 → 9:1 → 4:1) gave the crude Fmoc-protected compound (830 mg, 137%, mass balance) which was used without further analytics. The Fmoc-protected compound (830 mg, 0.94 mmol) was dissolved in 5 mL of a 20% piperidine solution in DMF and stirred at r.t. for 4 h. The amount of the DMF was reduced by evaporation, the residue was dissolved in EA, washed with water and brine and the organic phase dried over Na_2_SO_4_. After flash chromatography (silica, gradient DCM/MeOH: 100:0 → 99:1 → 95:5 → 90:10) the amine (240 mg, 60%) was used directly for the next reaction. To a solution of the amine (240 mg, 0.57 mmol, 1 eq.) and TEA (87 µL, 0.62 mmol, 1.1 eq.) in 4 mL DCM, 2-bromopropanoyl bromide (135 mg, 0.62 mmol, 1.1 eq.) was added. After stirring the mixture for 2.5 h, the solvent was evaporated and the crude product was purified by flash chromatography (silica, gradient PE/EA: 8:2 → 7:3 → 6:4 → 5:5) to give compound **9** (180 mg, 57%, 34% overall yield) as a white-beige solid.

^1^H-NMR (400 MHz, CDCl_3_) *δ* 9.38 (s, 1H), 7.89 (d, *J* = 1.8 Hz, 1H), 7.85–7.78 (m, 2H), 7.38–7.17 (m, 8H), 7.10 (t, *J* = 6.0 Hz, 1H), 4.52–4.33 (m, 3H), 1.85 (d, *J* = 7.0 Hz, 3H), 1.47 (s, 9H); ^13^C-NMR (101 MHz, CDCl_3_) *δ* 169.82, 165.83, 154.73, 141.99, 141.19, 133.45, 133.16, 130.75, 129.32, 127.93, 127.51, 126.34, 126.29, 124.93, 124.03, 123.51, 120.49, 81.41, 44.60, 43.54, 28.40, 22.89; HRFT-MS (ESI+): *m*/*z* = 580.0858 and 582.0850 (calc. 580.0876 and 582.0856 for [M + Na]^+^).

#### 3.3.26. Synthesis of *N*-[2-amino-5-(thiophen-2-yl)phenyl]-4-[(2-fluoropropanamido)methyl]benzamide (**BA1**)

Compound **4a** (140 mg, 0.28 mmol) was synthesized according to the general procedure **A4** to give after filtration compound **BA1** (53 mg, 53%) as light-brown solid.

UV purity: 98% (t_R_ = 8.69 min); ^1^H-NMR (400 MHz, DMSO-*d_6_*) *δ* 9.70 (s, 1H), 8.84–8.76 (m, 1H), 7.96 (d, *J* = 8.2 Hz, 2H), 7.48 (d, *J* = 2.2 Hz, 1H), 7.39 (d, *J* = 8.2 Hz, 2H), 7.35 (dd, *J* = 5.1, 1.2 Hz, 1H), 7.30 (dd, *J* = 8.3, 2.2 Hz, 1H), 7.24 (dd, *J* = 3.6, 1.2 Hz, 1H), 7.05 (dd, *J* = 5.1, 3.6 Hz, 1H), 6.82 (d, *J* = 8.3 Hz, 1H), 5.19–4.99 (m, 3H), 4.39 (d, *J* = 6.1 Hz, 2H), 1.47 (dd, *J* = 24.5, 6.7 Hz, 3H); ^19^F-NMR (377 MHz, DMSO-*d_6_*) *δ* −181.06 (^1^H decoupled); ^13^C-NMR (101 MHz, DMSO-*d_6_*) *δ* 169.99 (d, *J* = 20.2 Hz), 165.29, 144.23, 142.99, 142.82, 133.07, 128.20, 127.90, 126.90, 123.90, 123.40, 123.21, 122.24, 121.01, 116.36, 87.96 (d, *J* = 179.1 Hz), 41.50, 18.47 (d, *J* = 22.0 Hz); HRFT-MS (ESI+): *m*/*z* = 398.1341 (calc. 398.1333 for [M+H]^+^).

#### 3.3.27. Synthesis of *N*-[2-amino-5-(thiophen-2-yl)phenyl]-4-(2-fluoropropanamido)benzamide (**BA2**)

Compound **5a** (200 mg, 0.414 mmol) was synthesized according to the general procedure **A4** to give after filtration compound **BA2** (144 mg, 91%) as greyish solid.

UV purity: 99% (t_R_ = 8.95 min); ^1^H-NMR (400 MHz, DMSO-*d_6_*) *δ* 10.34 (s, 1H), 9.67 (s, 1H), 8.03–7.96 (m, 2H), 7.86–7.78 (m, 2H), 7.48 (d, *J* = 2.2 Hz, 1H), 7.35 (dd, *J* = 5.1, 1.1 Hz, 1H), 7.30 (dd, *J* = 8.3, 2.2 Hz, 1H), 7.25 (dd, *J* = 3.5, 1.2 Hz, 1H), 7.05 (dd, *J* = 5.1, 3.6 Hz, 1H), 6.82 (d, *J* = 8.3 Hz, 1H), 5.23 (dq, *J* = 48.5, 6.6 Hz, 1H), 5.14 (s, 2H), 1.55 (dd, *J* = 24.6, 6.7 Hz, 3H); ^19^F-NMR (377 MHz, DMSO) *δ* −179.76 (^1^H decoupled); ^13^C-NMR (101 MHz, DMSO-*d_6_*) *δ* 168.85 (d, *J* = 20.6 Hz), 164.78, 142.59, 141.69, 140.96, 129.69, 128.63, 126.54, 125.75, 124.57, 124.46, 123.89, 123.49, 119.17, 117.65, 116.33, 87.79 (d, *J* = 179.0 Hz), 18.28 (d, *J* = 22.2 Hz); HRFT-MS (ESI+): *m*/*z* = 384.1196 (calc. 384.1177 for [M+H]^+^).

#### 3.3.28. Synthesis of *N*-[2-amino-5-(thiophen-3-yl)phenyl]-4-[(2-fluoropropanamido)methyl]benzamide (**BA3**)

Compound **4b** (530 mg, 1.06 mmol) was synthesized according to the general procedure **A4** to give after filtration compound **BA3** (244 mg, 58%) as beige solid.

UV purity: 99% (t_R_ = 8.60 min); ^1^H-NMR (400 MHz, DMSO-*d_6_*) *δ* 9.71 (s, 1H), 8.80 (dt, *J* = 6.3, 3.3 Hz, 1H), 7.97 (d, *J* = 8.0 Hz, 2H), 7.60–7.50 (m, 3H), 7.45–7.33 (m, 4H), 6.82 (d, *J* = 8.3 Hz, 1H), 5.20–4.96 (m, 3H), 4.39 (d, *J* = 6.1 Hz, 2H), 1.47 (dd, *J* = 24.5, 6.7 Hz, 3H); ^19^F-NMR (377 MHz, DMSO-*d_6_*) *δ* −181.06 (dq, *J* = 49.0, 24.5 Hz); ^13^C-NMR (101 MHz, DMSO-*d_6_*) *δ* 169.99 (d, *J* = 20.3 Hz), 165.22, 142.76, 142.55, 141.68, 133.14, 127.86, 126.90, 126.55, 125.74, 124.54, 124.50, 123.88, 123.41, 117.66, 116.32, 87.96 (d, *J* = 179.0 Hz), 41.51, 18.47 (d, *J* = 21.9 Hz); HRFT-MS (ESI+): *m*/*z* = 420.1163 (calc. 420.1152 for [M+Na]^+^).

#### 3.3.29. Synthesis of *N*-[2-amino-5-(thiophen-3-yl)phenyl]-4-(2-fluoropropanamido)benzamide (**BA4**)

Compound **5b** (100 mg, 0.207 mmol) was synthesized according to the general procedure **A4** to give after filtration compound **BA4** (60 mg, 76%) as light-brown solid.

UV purity: 99% (t_R_ = 8.86 min); ^1^H-NMR (400 MHz, DMSO-*d_6_*) *δ* 10.34 (s, 1H), 9.67 (s, 1H), 8.04–7.97 (m, 2H), 7.87–7.80 (m, 2H), 7.49 (d, *J* = 2.2 Hz, 1H), 7.35 (dd, *J* = 5.1, 1.1 Hz, 1H), 7.30 (dd, *J* = 8.3, 2.2 Hz, 1H), 7.25 (dd, *J* = 3.6, 1.2 Hz, 1H), 7.05 (dd, *J* = 5.1, 3.6 Hz, 1H), 6.82 (d, *J* = 8.3 Hz, 1H), 5.32–5.09 (m, 3H), 1.56 (dd, *J* = 24.5, 6.7 Hz, 3H); ^19^F-NMR (377 MHz, DMSO-*d_6_*) *δ* −179.75 (^1^H decoupled); ^13^C-NMR (101 MHz, DMSO-*d_6_*) *δ* 168.86 (d, *J* = 20.5 Hz), 164.85, 144.25, 143.03, 141.00, 129.62, 128.67, 128.21, 123.92, 123.88, 123.48, 123.20, 122.27, 121.01, 119.18, 116.38, 87.80 (d, *J* = 179.1 Hz), 18.28 (d, *J* = 22.1 Hz); HRFT-MS (ESI+): *m*/*z* = 384.1184 (calc. 384.1177 for [M + H]^+^).

#### 3.3.30. Synthesis of *N*-(4-amino-4′-fluoro-[1,1′-biphenyl]-3-yl)-4-[(2-fluoropropanamido)methyl]benzamide (**BA5**)

Compound **4c** (255 mg, 0.4 mmol) was synthesized according to the general procedure **A4** to give after filtration compound **BA5** (104 mg, 64%) as white solid.

UV purity: 99% (t_R_ = 8.90 min); ^1^H-NMR (400 MHz, DMSO-*d_6_*) *δ* 9.70 (s, 1H), 8.80 (t, *J* = 6.4 Hz, 1H), 7.96 (d, *J* = 7.9 Hz, 2H), 7.58 (dd, *J* = 8.5, 5.4 Hz, 2H), 7.53–7.46 (m, 1H), 7.39 (d, *J* = 7.9 Hz, 2H), 7.30 (dd, *J* = 8.3, 2.3 Hz, 1H), 7.21 (t, *J* = 8.7 Hz, 2H), 6.86 (d, *J* = 8.3 Hz, 1H), 5.09 (s, 2H), 5.09 (dq, *J* = 49.0, 6.7 Hz, 1H), 4.39 (d, *J* = 6.1 Hz, 2H), 1.47 (dd, *J* = 24.5, 6.7 Hz, 3H); ^19^F-NMR (377 MHz, DMSO-*d_6_*) *δ* −117.39–−117.60 (m), −181.07 (ddd, *J* = 47.4, 28.4, 14.1 Hz); ^13^C-NMR (101 MHz, DMSO-*d_6_*) *δ* 170.00 (d, *J* = 20.3 Hz), 165.30, 161.01 (d, *J* = 242.6 Hz), 142.79, 142.70, 136.75, 133.18, 127.88, 127.30 (d, *J* = 7.7 Hz), 127.16, 126.92, 124.73, 124.63, 123.59, 116.51, 115.50 (d, *J* = 21.3 Hz), 87.97 (d, *J* = 179.3 Hz), 41.51, 18.48 (d, *J* = 22.1 Hz); HRFT-MS (ESI+): *m*/*z* = 410.1704 (calc. 410.1675 for [M + H]^+^).

#### 3.3.31. Synthesis of *N*-(4-amino-4′-fluoro-[1,1′-biphenyl]-3-yl)-4-(2-fluoropropanamido)benzamide (**BA6**)

Compound **5c** (160 mg, 0.323 mmol) was synthesized according to the general procedure **A4** to give after filtration compound **BA6** (86 mg, 67%) as white solid.

UV purity: 95% (t_R_ = 9.15 min); ^1^H-NMR (400 MHz, DMSO-*d_6_*) *δ* 10.34 (s, 1H), 9.67 (s, 1H), 8.03–7.97 (m, 2H), 7.83 (d, *J* = 8.7 Hz, 2H), 7.61–7.55 (m, 2H), 7.50 (d, *J* = 2.2 Hz, 1H), 7.30 (dd, *J* = 8.3, 2.3 Hz, 1H), 7.25–7.16 (m, 2H), 6.87 (d, *J* = 8.4 Hz, 1H), 5.23 (dq, *J* = 48.6, 6.7 Hz, 1H), 5.09 (s, 2H), 1.55 (dd, *J* = 24.5, 6.7 Hz, 3H); ^19^F-NMR (377 MHz, DMSO-*d_6_*) *δ* −117.50 (tt, *J* = 9.0, 5.3 Hz), −179.77 (dq, *J* = 48.9, 24.4 Hz); ^13^C-NMR (101 MHz, DMSO-*d_6_*) *δ* 168.85 (d, *J* = 20.6 Hz), 164.84, 161.01 (d, *J* = 242.7 Hz), 142.73, 140.97, 136.73 (d, *J* = 3.1 Hz), 129.71, 128.64, 127.30 (d, *J* = 7.9 Hz), 127.16, 124.76, 124.59, 123.66, 119.18, 116.51, 115.50 (d, *J* = 21.2 Hz), 87.79 (d, *J* = 179.1 Hz), 18.27 (d, *J* = 22.2 Hz); HRFT-MS (ESI+): *m*/*z* = 396.1540 (calc. 396.1518 for [M+H]^+^).

#### 3.3.32. Synthesis of *N*-[2-amino-5-(furan-2-yl)phenyl]-4-[(2-fluoropropanamido)methyl]benzamide (**BA7**)

Compound **4d** (260 mg, 0.54 mmol) was synthesized according to the general procedure **A4** to give after filtration compound **BA7** (39 mg, 19%) beige solid.

UV purity: 95% (t_R_ = 8.41 min); ^1^H-NMR (400 MHz, DMSO-*d_6_*) *δ* 9.67 (s, 1H), 8.84–8.76 (m, 1H), 7.99–7.93 (m, 2H), 7.60 (dd, *J* = 1.9, 0.8 Hz, 1H), 7.54 (d, *J* = 2.0 Hz, 1H), 7.39 (d, *J* = 8.0 Hz, 2H), 7.33 (dd, *J* = 8.3, 2.1 Hz, 1H), 6.82 (d, *J* = 8.4 Hz, 1H), 6.61 (dd, *J* = 3.3, 0.8 Hz, 1H), 6.51 (dd, *J* = 3.4, 1.8 Hz, 1H), 5.20–4.98 (m, 3H), 4.39 (d, *J* = 6.2 Hz, 2H), 1.47 (dd, *J* = 24.5, 6.7 Hz, 3H); ^19^F-NMR (377 MHz, DMSO-*d_6_*) *δ* −181.06 (dq, *J* = 49.1, 24.5 Hz); ^13^C-NMR (101 MHz, DMSO-*d_6_*) *δ* 169.99 (d, *J* = 20.3 Hz), 165.31, 153.79, 142.92, 142.79, 141.21, 133.12, 127.88, 126.95, 126.91, 123.24, 122.27, 119.18, 116.18, 111.78, 102.39, 87.96 (d, *J* = 179.1 Hz), 41.51, 18.47 (d, *J* = 22.0 Hz); HRFT-MS (ESI+): *m*/*z* = 382.1563 (calc. 382.1561 for [M+H]^+^).

#### 3.3.33. Synthesis of *N*-[2-amino-5-(furan-2-yl)phenyl]-4-(2-fluoropropanamido)benzamide (**BA8**)

Compound **5d** (240 mg, 0.513 mmol) was synthesized according to the general procedure **A4** to give after filtration compound **BA8** (29 g, 15%) as orange solid.

UV purity: 95% (t_R_ = 8.77 min); ^1^H-NMR (400 MHz, DMSO-*d_6_*) *δ* 10.35 (s, 1H), 9.65 (s, 1H), 8.03–7.95 (m, 2H), 7.86–7.79 (m, 2H), 7.60 (d, *J* = 1.8 Hz, 1H), 7.53 (d, *J* = 2.1 Hz, 1H), 7.33 (dd, *J* = 8.3, 2.1 Hz, 1H), 6.82 (d, *J* = 8.4 Hz, 1H), 6.61 (d, *J* = 3.4 Hz, 1H), 6.50 (dd, *J* = 3.3, 1.8 Hz, 1H), 5.34–5.09 (m, 3H), 1.55 (dd, *J* = 24.6, 6.7 Hz, 3H); ^19^F-NMR (377 MHz, DMSO-*d_6_*) *δ* −179.78 (dq, *J* = 48.9, 24.6 Hz); ^13^C-NMR (101 MHz, DMSO-*d_6_*) *δ* 168.84 (d, *J* = 20.5 Hz), 164.86, 153.80, 142.94, 141.19, 140.98, 129.66, 128.63, 123.33, 122.30, 122.20, 119.20, 119.16, 116.19, 111.78, 102.38, 87.77 (d, *J* = 179.1 Hz), 18.27 (d, *J* = 22.2 Hz); HRFT-MS (ESI+): *m*/*z* = 368.1408 (calc. 368.1405 for [M+H]^+^).

#### 3.3.34. Synthesis of *N*-[2-amino-5-(furan-3-yl)phenyl]-4-[(2-fluoropropanamido)methyl]benzamide (**BA9**)

Compound **4e** (234 mg, 0.486 mmol) was synthesized according to the general procedure **A4** to give after filtration compound **BA9** (50 mg, 27%) as beige solid.

UV purity: 96% (t_R_ = 8.24 min); ^1^H-NMR (400 MHz, DMSO-*d_6_*) *δ* 9.70 (s, 1H), 8.84–8.76 (m, 1H), 8.00–7.93 (m, 3H), 7.66 (t, *J* = 1.7 Hz, 1H), 7.42–7.35 (m, 3H), 7.24 (dd, *J* = 8.3, 2.1 Hz, 1H), 6.86–6.77 (m, 2H), 5.09 (dq, *J* = 48.9, 6.7 Hz, 1H), 4.97 (s, 2H), 4.39 (d, *J* = 6.1 Hz, 2H), 1.47 (dd, *J* = 24.5, 6.7 Hz, 3H); ^19^F-NMR (377 MHz, DMSO-*d_6_*) *δ* −181.05 (dq, *J* = 49.0, 24.5 Hz); ^13^C-NMR (101 MHz, DMSO-*d_6_*) *δ* 170.00 (d, *J* = 20.1 Hz), 165.19, 143.82, 142.76, 142.42, 137.34, 133.10, 127.84, 126.90, 125.93, 124.15, 124.09, 123.46, 120.22, 116.34, 108.53, 87.96 (d, *J* = 179.1 Hz), 41.50, 18.47 (d, *J* = 21.9 Hz); HRFT-MS (ESI+): *m*/*z* = 382.1564 (calc. 382.1561 for [M+H]^+^).

#### 3.3.35. Synthesis of *N*-[2-amino-5-(furan-3-yl)phenyl]-4-(2-fluoropropanamido)benzamide (**BA10**)

Compound **5e** (150 g, 0.32 mmol) was synthesized according to the general procedure **A4** to give after filtration compound **BA10** (104 mg, 88%) as light-brown solid.

UV purity: 96% (t_R_ = 8.49 min); ^1^H-NMR (400 MHz, DMSO-*d_6_*) *δ* 10.35 (s, 1H), 9.68 (s, 1H), 8.04–7.94 (m, 3H), 7.87–7.78 (m, 2H), 7.66 (t, *J* = 1.7 Hz, 1H), 7.40 (d, *J* = 2.1 Hz, 1H), 7.24 (dd, *J* = 8.2, 2.1 Hz, 1H), 6.84–6.77 (m, 2H), 5.23 (dq, *J* = 48.5, 6.7 Hz, 1H), 4.98 (s, 2H), 1.56 (dd, *J* = 24.5, 6.7 Hz, 3H); ^19^F-NMR (377 MHz, DMSO-*d_6_*) *δ* −179.77 (^1^H decoupled); ^13^C-NMR (101 MHz, DMSO-*d_6_*) *δ* 168.86 (d, *J* = 20.7 Hz), 164.75, 143.82, 142.47, 140.97, 137.35, 129.65, 128.62, 125.95, 124.20, 124.06, 123.54, 120.24, 119.17, 116.35, 108.54, 87.79 (d, *J* = 179.0 Hz), 18.28 (d, *J* = 22.1 Hz); HRFT-MS (ESI+): *m*/*z* = 368.1416 (calc. 368.1405 for [M+H]^+^).

### 3.4. Radiosynthesis

#### 3.4.1. Manual Radiosynthesis of [^18^F]**BA3**

No-carrier-added (n.c.a.) [^18^F]fluoride was produced via the ^18^O(p,n)^18^F nuclear reaction by irradiation of the conical shaped [^18^O]H_2_O (Hyox 18 enriched water; Rotem Industries Ltd., Mishor Yamin, Israel) target Nirta^®^ Conical 5 (IBA RadioPharma Solutions, Louvain-la-Neuve, Belgium) on a Cyclone 18/9 (IBA RadioPharma Solutions, Louvain-la-Neuve, Belgium) with 18 MeV proton beam or [^18^O]H_2_O recycled by the established in-house method [66]. [^18^F]Fluoride in 1.5 mL of H_2_O was trapped on a Sep-Pak^®^ Accell QMA light cartridge (sorbent weight: 46 mg; preconditioned with 10 mL of an aqueous 0.5 M NaHCO_3_ solution and 10 mL water, Waters, Milford, MA, USA). The activity was eluted with 390 µL of an aqueous solution of K_2_CO_3_ (1.8 mg, 13 μmol), KHCO_3_ (1.35 mg, 13 μmol) or K_2_C_2_O_4_ (1.84 mg, 10 μmol), respectively, into a 5 mL microwave V-vial (CEM^®^ Corporation, Matthews, NC, USA) and K_2.2.2_ (11 mg, 29 μmol) or 18-crown-6 (10 mg, 38 µmol), respectively, in 1 mL of MeCN was added. For investigations of the precursor to base ratio, the amount of potassium carbonate was accordingly reduced or increased. For labelling with [^18^F]TBAF, the tetra-*n*-butylammonium hydrogen carbonate (TBAHCO_3_) solution (150 µL, 0.075 M, ABX, Radeberg, Germany) was directly placed into the V-vial containing [^18^F]fluoride and 1 mL of MeCN. The solution was azeotropically dried under vacuum and argon flow in the microwave cavity (Discover PETwave microwave CEM^®^ corporation, 50–60 °C, 75 W) for 8–10 min. Additional aliquots of MeCN (2 × 1.0 mL) were added during the drying process and the final complex was dissolved in an appropriate volume of labelling solvent and used directly or divided in several portions. Thereafter, a solution containing the bromo precursor **9** (2–3 mg, 3.6–5.4 µmol) in 300 µL of an appropriate solvent was added, and ^18^F-labelling was performed at different temperatures in dependence of the solvent used (80, 100 and 150 °C). The reaction was monitored in different time points (up to 20 min) via radio-thin layer chromatography (radio-TLC) and (radio-)reversed phase high-performance liquid chromatography ((radio)-RP-HPLC, see quality control).

#### 3.4.2. Automated Radiosynthesis of [^18^F]**BA3**

Remotely controlled radiosynthesis was performed using a TRACERlab FX2 N synthesizer (GE Healthcare, Waukesha, WI, USA) equipped with a Laboport vacuum pump N810.3FT.18 (KNF Neuburger GmbH, Freiburg, Germany), a BlueShadow UV detector 10D (KNAUER GmbH, Berlin, Germany and the TRACERlab FX Software (Version 2.3.0, GE Healthcare, Waukesha, WI, USA) with the optimized radiolabelling conditions. [^18^F]TBAF complex was obtained after trapping [^18^F]fluoride (7–32 GBq) on a Sep-Pak^®^ Accell QMA light cartridge (Appendix A, entry 1, sorbent weight: 46 mg; preconditioned with 10 mL of an aqueous 0.5 M NaHCO_3_ solution and 10 mL water, Waters, Milford, MA, USA) and eluted into the reactor with a solution of 150 µL of TBAHCO_3_ (entry 2, 0.075 M ABX GmbH, Radeberg, Germany), 300 µL of H_2_O and 600 µL of MeCN. After azeotropic drying at 50 °C for 3 min, 2 mL MeCN (entry 3) were added and azeotropic drying was continued for further 3 min at 70 °C and 1 min at 40 °C. Thereafter, the nucleophilic aliphatic radiofluorination proceeded by adding the bromo precursor **9** (4 mg, 7.2 µmol) dissolved in 800 µL of MeCN (entry 4) and stirring the reaction mixture at 100 °C for 15 min. The reaction mixture was cooled to 40 °C and the Boc deprotection was carried out by adding 800 µL of a 2 M HCl_aq_ solution (entry 5) and subsequent heating at 80 °C for another 5 min. After neutralization with a mixture of 1.6 mL of an aqueous 1 M NaHCO_3_ solution and 1.8 mL of 100 mM aqueous phosphate buffer (pH = 6, entry 6), the solution was transferred into the injection vial (entry 7). [^18^F]**BA3** was isolated by semipreparative RP-HPLC (column: Reprosil-Pur C18-AQ, 250 × 10 mm, 10 µm; Dr. Maisch HPLC GmbH, Ammerbuch, Germany) with a solvent composition of 40% MeCN/20 mM NH_4_OAc_aq_ at a flow rate of 4.0 mL/min (entry 8). The radiotracer [^18^F]**BA3** was collected in the HPLC collection vial containing 40 mL of H_2_O (entry 9), trapped on a Sep-Pak^®^ C18 light cartridge (entry 10, sorbent weight: 130 mg; preconditioned with 5 mL EtOH and 10 mL water, Waters, Milford, MA, USA) and followed by washing with 2 mL of H_2_O (entry 11) and elution of [^18^F]**BA3** with 1.3 mL EtOH (entry 12) in the product vial (entry 13). The ethanolic solution was transferred outside of the hotcell by remote control, the solvent was reduced manually in a stream of argon at 70 °C and the desired radiotracer was reconstituted in a saline solution (NaCl 0.9%) containing max. 10% EtOH (*v*/*v*). [^18^F]**BA3** was then ready for further biological characterization in a total synthesis time of about 85 min.

#### 3.4.3. Quality Control and Analyses

Radio-thin layer chromatography (radio-TLC) of [^18^F]**BA3** was performed on Alugram^®^ SIL G/UV254 pre-coated plates (Macherey-Nagel, Düren, Germany) with DCM/MeOH (4:1, *v*/*v*). The plates were exposed to storage phosphor screens (BAS IP MS 2025 E, GE Healthcare Europe GmbH, Freiburg, Germany) and recorded using the Amersham Typhoon RGB Biomolecular Imager (GE Healthcare Life Sciences, Freiburg, Germany). Images were quantified with the ImageQuant TL8.1 software (Version 8.1, GE Healthcare Life Sciences, Freiburg, Germany). Analytical chromatographic separations were performed on a JASCO LC-2000 system, incorporating a PU-2080Plus pump, AS-2055Plus auto injector (100 µL sample loop), and a UV-2070Plus detector (JASCO Deutschland GmbH, Pfungstadt, Germany) coupled with a gamma radioactivity HPLC flow detector (Gabi Star, raytest Isotopenmessgeräte GmbH, Straubenhardt, Germany). Data analysis was performed with the Galaxie chromatography software (Agilent Technologies, Santa Clara, CA, USA) using the chromatograms obtained at 254 nm. Radiochemical yield from aliquots of the reaction mixture, radiochemical purities and in vivo metabolism analysis of plasma, brain and urine samples were assessed via reverse phase—high performance liquid chromatography (RP-HPLC) with a Reprosil-Pur C18-AQ column (250 × 4.6 mm, 5 μm) in gradient mode (0–5 min: 10% MeCN/20 mM NH_4_OAc_aq_, 5–25 min: 10% → 90% MeCN/20 mM NH_4_OAc_aq_, 25–29 min: 90% MeCN/20 mM NH_4_OAc_aq_, 29–30 min: 90% → 10% MeCN/20 mM NH_4_OAc_aq_, 30–35 min: 10% MeCN/20 mM NH_4_OAc_aq_) and a flow of 1.0 mL/min.

The molar activity was determined using analytical RP-HPLC with a Reprosil-Pur C18-AQ column (250 × 4.6 mm, 5 μm) and 38% MeCN/20 mM NH_4_OAc_aq_ as eluent at a flow rate of 1.0 mL/min obtained at 254 nm.

The ammonium acetate concentrations stated as 20 mM NH_4_OAc_aq_ correspond to the concentration in the aqueous component of an eluent mixture.

#### 3.4.4. Determination of Radiochemical Stability and Lipophilicity (logD_7.4_)

Radiochemical stability of [^18^F]**BA3** was investigated in 0.9% NaCl solution (containing 10% EtOH) and in *n*-octanol at room temperature for 2 h. Samples were analysed by radio-RP-HPLC (see quality control). The LogD_7.4_ value of [^18^F]**BA3** was experimentally determined in *n*-octanol/phosphate-buffered saline (PBS, pH = 7.4) at room temperature by the shake-flask method as described previously [67]. Briefly, small tracer amounts (~800 kBq) were added to a mixture of 3 mL *n*-octanol and 3 mL PBS. After shaking for 20 min, the samples were centrifuged (10,000 rpm, 5 min) and 1 mL aliquots of the organic as well as aqueous layer were taken and measured in a gamma counter (1480 WIZARD, Perkin Elmer, Turku, Finland). Another 1 mL aliquot of the organic layer was mixed with 2 mL *n*-octanol and 3 mL PBS and subjected to the same procedure until constant partition coefficient values had been obtained. The measurement was performed twice in quadruplicate.

#### 3.4.5. Inhibition Assay for HDAC1-3 and HDAC6

In vitro inhibitory activities against HDAC1−3 and HDAC6 were measured as previously published [68] with minor modifications. In short, 3-fold serial dilutions of test compounds and controls in assay buffer (50 mM Tris-HCl, pH = 8.0, 137 mM NaCl, 2.7 mM KCl, 1.0 mM MgCl_2_ × 6 H_2_O, 0.1 mg/mL BSA) were prepared, and 5.0 µL were transferred into black microplates (OptiPlate-96, PerkinElmer Inc., Waltham, MA, USA). 25 µL of assay buffer and 10 µL enzyme solution (human recombinant HDAC1 (BPS Bioscience Inc., San Diego, CA, USA, Catalog# 50051); HDAC2 (BPS Bioscience Inc., San Diego, CA, USA, Catalog# 50052); HDAC3/NcoR2 (BPS Bioscience Inc., San Diego, CA, USA, Catalog# 50003); HDAC6 (BPS Bioscience Inc., San Diego, CA, USA, Catalog# 50006) were added. Enzyme and inhibitor were preincubated at 25 °C for 60 min (HDAC1-3) or 15 min (HDAC6). Afterwards, the fluorogenic substrate Z-Lys(Ac)-AMC (ZMAL) [69] (10 µL; 75 µM in assay buffer) was added. The total assay volume of 50 µL (max. 1% DMSO) was incubated at 37 °C for 90 min. Subsequently, 50 µL trypsin solution (0.4 mg/mL trypsin in buffer: 50 mM Tris−HCl, pH = 8.0, 100 mM NaCl) was added, followed by additional 30 min of incubation at 37 °C. Fluorescence (excitation: 355 nm, emission: 460 nm) was measured using a Thermo Scientific^TM^ Ascent Fluoroskan^TM^ microplate reader (Thermo Fisher Scientific, Waltham, MA, USA). All compounds were tested at least twice in duplicates. The 50% inhibitory concentration (IC_50_) was determined by plotting the dose response curves and nonlinear regression with GraphPad Prism (GraphPad Prism 9.3 for MacOS, GraphPad Software, LLC., San Diego, CA, USA).

### 3.5. Biological Experiments

All experimental work including animals has been conducted in accordance with the national legislation on the use of animals for research (Tierschutzgesetz [TierSchG], Tierschutz-Versuchstierverordnung [TierSchVersV]) and were approved by the responsible authorities of Saxony (Landesdirektion Sachsen, Referat 25—Veterinärwesen und Lebensmittelüberwachung; TVV 18/18, DD24.1-5131/446/19, valid until 23 June 2023). CD-1 mice (female, 8–15 weeks, 29–43 g) were obtained from the Medizinisch-Experimentelles Zentrum at the Medical Faculty of the Leipzig University (Leipzig, Germany).

#### 3.5.1. Metabolite Analysis

The radiotracer [^18^F]**BA3** (~30 MBq in 200 µL isotonic saline) was administered via the tail vein in non-anesthetized female CD-1 mice (*n* = 2). At 30 min p.i., the animals were anesthetized by isoflurane inhalation and blood was sampled from the retro-orbital plexus. After cervical dislocation, urine was obtained and the brain was isolated. Plasma was obtained by centrifugation of the blood sample (8000× *g*, 2 min), and the brain was homogenized in ~1 mL distilled water on ice using a glass/PTFE Potter Homogenizer (Potter S Homogenizer, B. Braun Melsungen AG, Melsungen, Germany).

Protein precipitation was performed by addition of an ice-cold MeCN/H_2_O mixture (9:1, *v*/*v*) in a ratio of 4:1 (*v*/*v*) of organic solvent to plasma or brain homogenate, respectively. The samples were vortexed for 3 min, equilibrated on ice for 3 min, and centrifuged for 10 min at 10,000 rpm. The precipitates were washed with 100 μL of the solvent mixture and subjected to the same procedure. The combined supernatants were concentrated at 70 °C under argon flow to a final volume of approximately 100 μL and analysed by analytical radio-RP-HPLC with a gradient system (see quality control section). The peak corresponding to the radiotracer in the radio-chromatogram was identified by co-injecting the sample with the reference compound **BA3**. To determine the percentage of activity in the supernatants compared with total activity, aliquots of each step as well as the precipitates were quantified by an automated gamma counter (1480 WIZARD, Perkin Elmer, Turku, Finland). For plasma and brain samples recoveries of 97% and 95%, respectively, of total activity were obtained.

For the preparation of the MLC samples, 20 µL of mouse plasma was dissolved in 200 µL of an aqueous 200 mM sodium dodecyl sulfate (SDS_aq_) solution and injected into the MLC system. For brain samples, 400 µL brain homogenate was dissolved in 800 µL 200 mM SDS_aq_, heated for 5 min at 70 °C and centrifuged for 5 min at 10,000 rpm. The MLC system was built up as earlier described [61]. Separations were performed by using a Reprosil-Pur C18-AQ column (250 × 4.6 mm, 10 µm + 10 mm pre-column) and an eluent mixture of EtOH/100 mM SDS_aq_/25 mM (NH_4_)_2_HPO_4_ in gradient mode (0–10 min at 100% 100 mM SDS_aq_, 10–15 min up to 10% EtOH, 15–20 min at 10% EtOH, 20–21 min up to 100% 100 mM SDS_aq_; 21–30 min at 100% 100 mM SDS_aq_) at a flow rate of 1.0 mL/min.

The sodium dodecyl sulfate concentrations stated as 100 mM SDS_aq_ correspond to the concentration in the aqueous component of an eluent mixture.

#### 3.5.2. Cell Culture

F98 cells (obtained from ATCC^®^ Washington, DC, USA) and U251-MG cells (obtained from Drs A. Feldmann and N. Metwasi, Department of Radioimmunology, HZDR, Rossendorf, Germany) were cultivated in Dulbecco’s Modified Eagle Medium (DMEM, Gibco, Invitrogen, Dun Laoghaire, Ireland) supplemented with 10% heat inactivated fetal bovine serum (Gibco, Invitrogen, Dun Laoghaire, Ireland), and 100 U/mL penicillin and 100 µg/mL streptomycin (Gibco, Invitrogen, Dun Laoghaire, Ireland) in a humidified atmosphere in a CO_2_ incubator (37 °C, 5% CO_2_).

#### 3.5.3. Immunofluorescence Staining

Cells were seeded at 5 × 10^4^/well in an 8 well micro-slide (Ibidi GmbH, Gräfeling, Germany) 48 h before staining. Cells were fixed in 2% PFA (paraformaldehyde) for 20 min at room temperature and then washed twice with PBS. After a 45 min permeabilization/blocking step (0.3 Triton X 100, 5% normal goat serum in PBS), the detection of the HDAC1 was performed by overnight incubation at 4 °C of the slides with the primary rabbit polyclonal antibody (1:500 in blocking buffer 5% normal goat serum, GTX100513, Biozol Diagnostica Vertrieb GmbH, Eching, Germany). After washing with a solution of 1% BSA (bovine serum albumin) in PBS, the slides were incubated for 1 h at room temperature with the goat polyclonal secondary antibody to rabbit IgG (1:200 in dilution buffer 1% BSA, Alexa Fluor^®^ 488; ab150081, Abcam, Berlin, Germany). To finish, cells nuclei were counterstained with Hoechst, 10 min at room temperature (1:1000 in PBS, Hoechst 33258, Life Technologies, Carlsbad, CA, USA). After a step of washing and drying, slides were covered up with mounting medium (Aquapolymount, Polysciences Europe GmbH, Hirschberg an der Bergstrasse, Germany) for observation by fluorescence microscopy (Leica, DMi8, software Leica LASX, Leica Mikrosysteme Vertrieb GmbH, Wetzlar, Germany).

#### 3.5.4. MTS Assay

Proliferation of F98 and U251-MG cells was determined using a commercially available MTS assay (CellTiter 96^®^ AQueous One Solution Cell Proliferation Assay, Promega GmbH, Walldorf, Germany). Briefly, the cells were seeded into 96-well plates at a density of 5 × 10^4^ cells/well 24 h prior to the experiment at 37 °C in a humidified, 5% CO_2_ atmosphere. On the day of the experiment, the cell culture medium was replaced by fresh medium supplemented with either test compounds (working concentration: 50 µM; prepared by 1:200 dilution from a 10 mM stock solution in 100% DMSO with cell culture medium) or vehicle (0.5% DMSO in cell culture medium), and the cells were incubated for 72 h. Subsequently, 20 µL of the MTS solution was added per well and the plate was incubated in the CO_2_ incubator for another 2 h. Afterwards, the absorbance was measured at a wavelength of 490 nm using a microplate reader (Model 680 microplate reader, Bio-Rad Laboratories GmbH, Feldkirchen, Germany).

#### 3.5.5. PET Studies

For the time of the experiments, female CD-1 mice (*n* = 6; 8–15 weeks; 29–43 g) were kept in a dedicated climatic chamber with free access to water and food under a 12:12 h dark:light cycle at a constant temperature (24 °C). The animals were anaesthetized (Anaesthesia Unit U-410, agntho’s, Lidingö, Sweden) with isoflurane (2.0%, 300 mL/min) delivered in a 60% oxygen/40% air mixture (Gas Blender 100 Series, MCQ instruments, Rome, Italy) and their body temperature maintained at 37 °C with a thermal bed system. For baseline studies [^18^F]**BA3** was injected into the tail vein (*n* = 2; 6.0 ± 0.6 MBq in 150 µL isotonic saline; 9.2 ± 0.7 nmol/kg; A_m_: 21 GBq/μmol, EOS) followed by a dynamic 60 min PET/MR scan (Mediso nanoScan^®^, Budapest, Hungary). Permeability-glycoprotein efflux transporter studies consisted in pre-treatment via i.v. injection of **cyclosporine A** (50 mg/kg in NaCl/DMSO/kolliphor, 7:1:2, *v*/*v*; *n* = 2) or of vehicle (NaCl/DMSO/kolliphor, 7:1:2, *v*/*v*; *n* = 2) 30 min prior to [^18^F]**BA3** (6.0 ± 0.9 MBq; 4.9 ± 0.7 nmol/kg; A_m_: 32–37 GBq/μmol, EOS). Each PET image was corrected for random coincidences, dead time, scatter and attenuation (AC), based on a whole body (WB) MR scan. The reconstruction parameters for the list mode data were 3D-ordered subset expectation maximization (OSEM), 4 iterations, 6 subsets, energy window: 400–600 keV, coincidence mode: 1–5, ring difference: 81. The mice were positioned prone in a special mouse bed (heated up to 37 °C), with the head fixed to a mouth piece for the anaesthetic gas supply with isoflurane in 40% air and 60% oxygen. The PET data were collected by a continuous WB scan during the entire investigation. Following the 60 min PET scan a T1 weighted WB gradient echo sequence (TR/TE: 20/6.4 ms, NEX: 1, FA: 25, FOV: 64 × 64 mm, Matrix: 128 × 128, STh: 0.5 mm) was performed for AC and anatomical orientation. Image registration and evaluation of the region of interest (ROI) was carried out with PMOD (PMOD Technologies LLC, v. 3.9, Zurich, Switzerland). The respective brain regions were identified using the mouse brain atlas template Ma-Benveniste-Mirrione-FDG. The activity data are expressed as mean SUV of the overall ROI.

## 4. Conclusions

A series of novel fluorinated benzamide-based HDAC inhibitors with high inhibition potency and specificity to HDAC1 and HDAC2 was synthesized. With IC_50_ values in the low nanomolar range against HDAC1 (4.8 nM each) and HDAC2 (64.3 and 39.9 nM, respectively), the inhibitors **BA1** and **BA3** emerged as the most promising compounds of the series. **BA3**, the most potent inhibitor of HDAC1 and HDAC2 was subsequently selected for radiofluorination. After optimizing the radiolabelling conditions, a two-step one-pot automated radiosynthesis of [^18^F]**BA3** using the [^18^F]TBAF-complex and the TRACERlab FX2 N radiosynthesizer was successfully established. [^18^F]**BA3** was finally obtained in a low radiochemical yield, but with a high radiochemical purity and good molar activities. The biological evaluation in mice revealed a low brain uptake along with a high amount of radiometabolites of [^18^F]**BA3** in the brain. These results led to the conclusion that the radiotracer [^18^F]**BA3** is not suitable for imaging HDAC1/2 in the brain. However, the pharmacological potency of **BA3** in vitro is comparable to that of HDAC inhibitors possessing clinical efficacy in the treatment of cancer diseases, making **BA3** a potential lead structure for the development of new HDAC inhibitors. Consequently, the scaffold of **BA3** requires further modifications to optimize properties such as the metabolic stability and the brain penetrance.

## Data Availability

Data is contained within the article or Supplementary Materials.

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
