# Peer review of "Development and Biological Evaluation of the First Highly Potent and Specific Benzamide-Based Radiotracer [18F]BA3 for Imaging of Histone Deacetylases 1 and 2 in Brain"

_pharmaceuticals, 2022, doi:10.3390/ph15030324_

Round 1

Reviewer 1 Report

In the manuscript entitled “Development and biological evaluation of the first highly po-tent and specific benzamide-based radiotracer [18F]BA3 for imaging of histone deacetylase 1 in brain” Clauß et al. describe the attempt to develop a new radiotracer for an important enzyme involved in epigenetic processes. Based on existing knowledge in the field they selected a lead with a non-hydroxamate warhead and successfully identified potent and selective structures. In a second step, the most promising compound was used as the starting point for the design and synthesis of a new radiotracer. The work presented consists of a huge amount of experimental data.

Without a doubt, this endeavour is of very high merit and the syntheses described are state-of-the-art, the design of the compound BA3 has one important flaw leading to ethical concerns. While I am not an expert in the field of RRR, I am not totally convinced, that the animal studies were justified at this stage of the program. The biological evaluation in mice revealed a high amount of brain-penetrable radiometabolites and, since the radiotracer is a substrate of the P-gp, a low brain uptake. Shouldn’t the authors have looked into this prior to animal studies? The fact, that the compound BA3 might be a P-gp substrate was unknown, maybe, but could have been predicted using freely available tools such as SwissADME. With such a prediction in hand, the authors should have aimed to clarify this issue in vitro before they performed in vivo experiments, I guess, but I have to admit that such ethical issues are not my field of expertise. Maybe they have performed some tests in this direction but I overlooked something important in the in vitro experiments described.

Besides this ethical question, there are some minor points to be addressed. Some passages of the introduction sound more like a scholarly text than an original paper. The expression „hydroxamic acids and benzamides (also known as ortho-aminoanilides)“ in the sentence „Over the years hydroxamic acids and benzamides (also known as ortho-aminoanilides) have emerged as the most frequently used ZBGs“ sounds odd. I would suggest to omit the insert „(also known as ortho-aminoanilides)“.

Likewise, the following phrase should be modified „Additionally, those so-called pan-HDAC inhibitors lack se-lectivity towards specific HDAC isoforms or classes.“ I would suggest: „Characterized as pan-HDAC inhibitors, these compounds notoriously lack selectivity towards specific HDAC isoforms or classes.“

What is „[11C]I“ in line 97?

The IUPAC names do not obey the recommendations for the nesting order of enclosing marks and might have been generated automatically without the necessary editing by a trained chemist. For example the name tert-butyl (2-(4-(2-fluoropropanamido)benzamido)-4-(thiophen-3-yl)phenyl)carbamate (5b) should be corrected to tert-butyl {2-[4-(2-fluoropropanamido)benzamido]-4-(thiophen-3-yl)phenyl}carbamate (5b), I suppose.

Reviewer 2 Report

This manuscript describes development and biological evaluation of a potent benzamide-based radioligand for PET imaging of HDAC1 in the brain. After the evaluation of in vitro inhibitory activities against HDAC isoforms using the para-substituted aminophenyl benzamide library, the authors selected BA3 as the most promising candidate as a PET probe. [18F]BA3 was radiosynthesized successfully, but the radiochemical yield was low. The biological evaluation of [18F]BA3 has demonstrated its low brain permeability in mice, suggesting that this radioligand is not suitable for imaging HDAC1 in the brain and needs further optimization.

The manuscript is well written. I agree with that BA3 is a good lead compound for the future development of HDAC inhibitors and think this study contributes significantly to the development of PET imaging probes with the unique HDAC1 selectivity.

Comments on the manuscript are as follows:

Major points

  1. The authors have introduced a 2-fluoropropanamide moiety to the benzamide compounds for radiolabeling. I think it is informative if the rationale for this choice of radiolabeling group is mentioned in the manuscript.
  2. The authors have stated that “BA3 was selected for radiolabelling as the most active inhibitor of HDAC1 and HDAC2” (page 5, line 175), but the manuscript title and abstract imply that BA3 is an HDAC1 radiotracer. They should discuss whether the BA3’s HDAC1 selectivity over HDAC2 (approx. 8.3-fold) is sufficient for selective in vivo HDAC1 imaging or not.
  3. The HDAC1 selectivity over HDAC2 of BA1 seems to be higher than that of BA3 (13.4-fold vs 8.3-fold). I am wondering why BA3 was selected for radiolabelling as an HDAC1 radiotracer, instead of BA1.

Minor points

  1. (Page 8, line 243) I suggest adding an analytical UV chromatogram of no-carrier-added formulated [18F]BA3 in Figure 2.
  2. (Page 20, line 766) I think “4b” is correct, not “4a”.
  3. (Page 23, line 891) The sorbent weight and preconditioning method for the QMA cartridge used should be specified.
  4. (Page 23, line 895) I think “aliphatic” is correct, not “aromatic”.
  5. (Page 23, line 906) The preconditioning method for the C18 cartridge used should be specified.
  6. (Page 23, line 968) The information (e.g. company name, version) about “GraphPad Prism” should be added.
  7. (Page 25, line 989) I suggest specifying radioactivity extraction efficiencies from the plasma and brain samples.
  8. (Page 26, line 1084) “brain-penetrable radiometabolites” It is unclear whether the radiometabolites are brain-penetrant or not, as the authors have mentioned in Results and discussion (Page 10, line 302). I suggest modifying the description.
  9. (Page 26, line 1084) “the radiotracer is a substrate of the P-gp” I think it is difficult to conclude so without confirming that the increased radioactivity in the brain by CsA treatment is derived from unchanged [18F]BA3. It cannot be ruled out that radiometabolites are contributing to the observed increase of brain radioactivity.
  10. (Figure S5) The difference between A and B is unclear. How about adding an MR-only image?
  11. (Figure S10) The chemical structure of BA4 is inserted in the NMR spectrum for BA2.
